# Faster Atlantic currents drive poleward expansion of temperate phytoplankton in the Arctic Ocean

L. Oziel [1,2,7 ✉], A. Baudena[3,7], M. Ardyna[4,5], P. Massicotte[2], A. Randelhoff[2], J.-B. Sallée[3], R. B. Ingvaldsen[6], E. Devred [1] & M. Babin[2]

The Arctic marine biome, shrinking with increasing temperature and receding sea-ice cover, is tightly connected to lower latitudes through the North Atlantic. By flowing northward through the European Arctic Corridor (the main Arctic gateway where 80% of in- and outflow takes place), the North Atlantic Waters transport most of the ocean heat, but also nutrients and planktonic organisms toward the Arctic Ocean. Using satellite-derived altimetry observations, we reveal an increase, up to two-fold, in North Atlantic current surface velocities over the last 24 years. More importantly, we show evidence that the North Atlantic current and its variability shape the spatial distribution of the coccolithophore *Emiliania huxleyi* (*Ehux*), a tracer for temperate ecosystems. We further demonstrate that bio-advection, rather than water temperature as previously assumed, is a major mechanism responsible for the recent poleward intrusions of southern species like *Ehux*. Our findings confirm the biological and physical "Atlantification" of the Arctic Ocean with potential alterations of the Arctic marine food web and biogeochemical cycles.

[1] Ocean and Ecosystem Sciences Division, Bedford Institute of Oceanography, Fisheries and Oceans Canada, P.O. Box 1006, Dartmouth, NS B2Y 4A2, Canada. [2] Takuvik International Research Laboratory, Université Laval (Canada) - CNRS (France), Département de biologie et Québec-Océan, Université Laval, Québec, QC G1V 0A6, Canada. [3] Sorbonne Université, LOCEAN-IPSL, CNRS/IRD/MNHN, 4 Place Jussieu, Paris 75005, France. [4] Sorbonne Université & CNRS, Laboratoire d'Océanographie de Villefranche-sur-Mer (LOV), 181 chemin du Lazaret, Villefranche-sur-mer 06300, France. [5] Department of Earth System Science, Stanford University, Stanford, CA 94305, USA. [6] Institute of Marine Research, N5817 Bergen, Norway. [7] Present address: Sorbonne Université & CNRS, Laboratoire d'Océanographie de Villefranche-sur-Mer (LOV), 181 chemin du Lazaret, Villefranche-sur-mer 06300, France. ✉email: laurent.oziel@obs-vlfr.fr

The European Arctic Corridor (EAC, Fig. 1) is the main Arctic gateway where 80% of both in- and outflow takes place[1]. One of the most prominent synoptic feature of the EAC is the northward flowing North Atlantic Waters (NAW) entering the Arctic Ocean through two main branches (indicated in red in Fig. 1), which separate around 70°N. One branch keeps flowing northward toward Fram Strait while the second one turns eastward into the Barents Sea (BS). Over the last three decades, in a context of amplified warming[2] and sea-ice loss[3], substantial changes have been documented in the NAW inflow with a two-fold increase of its volume occupation and a northward shift of the polar front structure in the BS[4]. The NAW inflow largely controls the physical and sea-ice conditions of the region[5–8]. In addition, almost 50% of the total Arctic primary production takes place in the EAC[1,9–11], which is of major importance for fisheries[12].

Recent warming of the EAC related to the NAW inflow has been suspected to trigger poleward intrusions of temperate phytoplankton[13–16] and species from higher trophic levels[17–21]. By carrying biomass and nutrients produced elsewhere, bioadvection has recently been proposed as an "essential mechanism" for ecosystem dynamics in the Arctic Ocean[22,23]. Although the actual role of advection has already been identified as a potential driver altering zooplankton dynamics[24–26], it has never been assessed quantitatively from observations for phytoplankton in the EAC, or more generally in the Arctic Ocean[22,23,27–29].

Here we investigate how ocean currents control the spatial dynamics of a specific coccolithophore bloom-forming species, Emiliania huxleyi (hereafter referred to as Ehux). Ehux is usually associated with the surface layer of the temperate NAW in summer, typically in a post-spring bloom context characterized by low nutrients, low light and strong stratification[30–32]. Since Ehux does not form winter resting spores, this tracer of Atlantic ecosystem is generally considered to be a summer visitor in the BS, unlike neritic diatoms species[13]. A combination of bottom-up (i.e., winter darkness, cold temperatures and intense vertical mixing; see Supplementary Note 2.6) and top-down controls (i.e., grazing, viral lysis[33–35]) prevent Ehux from year-to-year survival in the north-easternmost parts of the BS. Because of their high abundance in Ehux (i.e., $115 \times 10^6$ cells L$^{-1}$ [36]), coastal regions and fjords of the Norwegian Sea have been suspected to be the source of Ehux for the whole EAC[37].

Using a Lagrangian tracking approach based on satellite-derived current fields, we explore how the advection of Ehux cells from these upstream coastal temperate regions shapes the distribution of the massive Ehux blooms in the BS (i.e., via the "seeding effect"[38,39]). By combining satellite-derived altimetry with ocean-color Particulate Inorganic Carbon (PIC, a coccolithophore biomass proxy) estimates, we demonstrate a major role of bio-advection in phytoplankton transport along the EAC.

## Results and discussion

**Climatic modes and increasing NAW surface current velocities.** To document the interannual and decadal variability of surface NAW currents, we first performed an Empirical Orthogonal Function (EOF) analysis of the Sea Level Anomalies (1993–2016, see Material and Methods), on which the seasonal signal was removed. The EOF analysis showed that the two first modes of variability accounted for more than 80% of the non-seasonal variability (Fig. 2a; Supplementary Note 2.3). When summing the two first modes of variability, a dipole structure emerged, centered on the NAW path between the Barents and Norwegian Sea shelves (about east of 5°E) and the center of the Norwegian Sea (about west of 5°E, Fig. 2a). The associated time-series showed a linear positive trend of about +9 (normalized units, Fig. 2b) over the last 24 years with high energy at the interannual and decadal time scale (Supplementary Fig. 7). This trend toward a more positive phase corresponds to an increase of the sea level gradient across the NAW path, hence intensifying NAW currents, while negative phases are associated with weaker NAW currents.

To reveal the impact of changing sea level on surface velocity fields, absolute surface geostrophic velocities, derived from

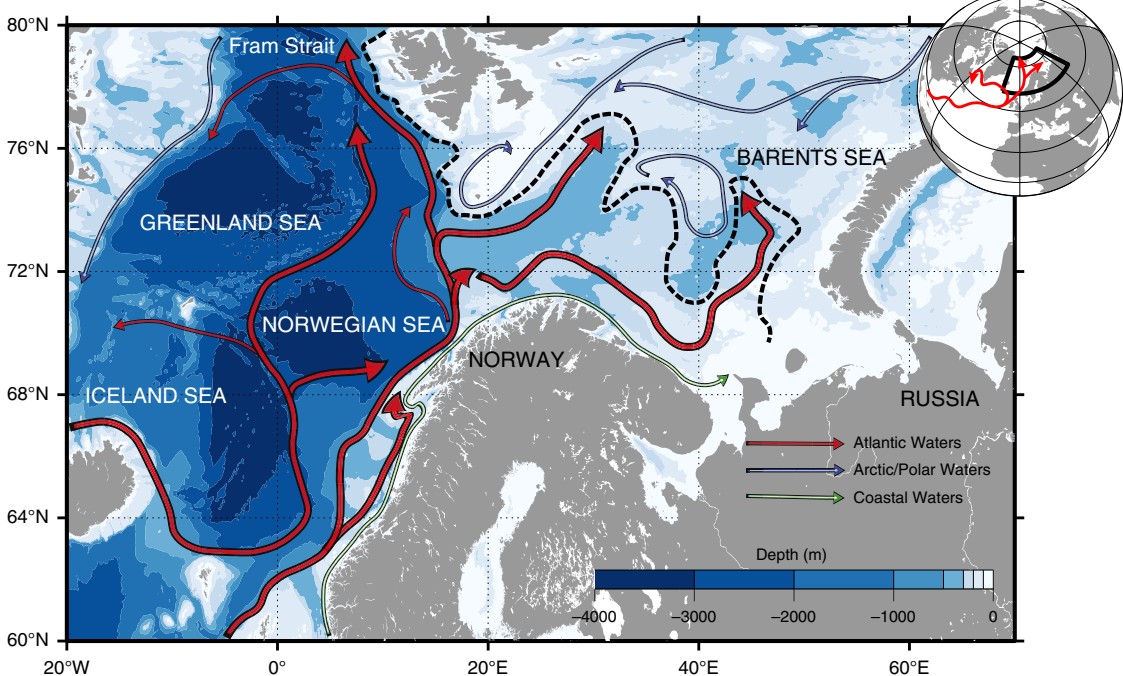

**Fig. 1 The European Arctic Corridor (EAC).** Bathymetry and surface circulation. The Atlantic currents are in red, the Arctic or Polar Waters are in blue and the Coastal Waters are in green. The southern Barents Sea Polar Front[4] is illustrated in black dashed line and separates the Atlantic Waters from the colder and fresher waters from the North.

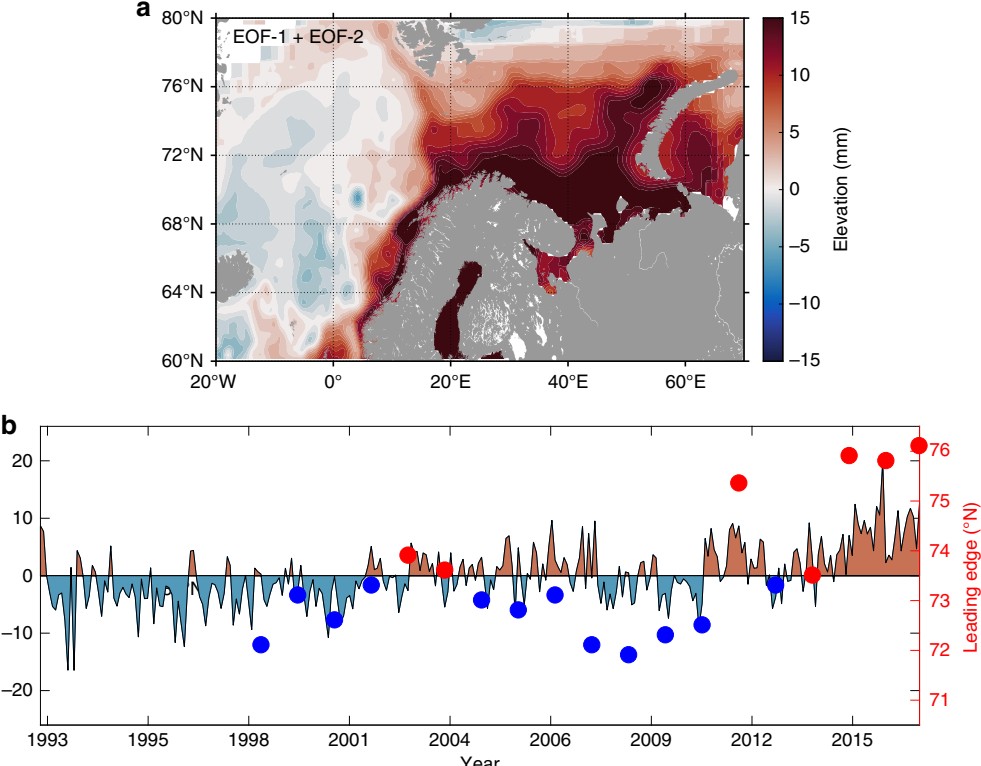

**Fig. 2 EOF analysis of the MSLA in the European Arctic Corridor.** The sum of the first two modes (**a**) account for more than 80% of the total variability. The associated time-series (**b**) experiences a positive linear trend. The blue/red dots correspond to the latitude of the leading-edge of the *Ehux* blooms in the eastern Barents Sea for longitudes > 40°E (right red y-axis), which strongly correlates with the yearly averaged EOF-1+EOF-2 time-series (r = 0.77, p = 0.0002, two-sided t-test).

Absolute Dynamic Topography fields, are shown in Fig. 3a, b during the extreme negative (i.e., 1993) and positive (i.e., 2015) phases of the EOF. These two contrasting years illustrate the ongoing strengthening of the ocean surface circulation in the EAC associated with the Atlantic inflow. Surface geostrophic current linear trends showed that surface currents changed in most of the Atlantic pathway (Fig. 3c). In the Norwegian Sea, the increase in the Atlantic surface current speed reached +2 mm s$^{-1}$ yr$^{-1}$, corresponding to a relative increase ranging from 30% on the shelf to 100% in the basin. The surface currents in the eastern BS Atlantic corridor also significantly increased by about +0.6 mm s$^{-1}$ yr$^{-1}$ (+14%). The positive EOF phase associated with stronger north-eastward NAW surface currents since year 2000 reflects an increase in advection at or near the surface. This result is consistent with the recent trends in NAW current velocities observed upstream in the Nordic Seas[40] or modelled in the BS[41].

The interannual fluctuations as well as the long-term trend observed in the time-series of surface geostrophic velocities are mainly attributed to the dynamics of the North Atlantic Subpolar Gyre (Pearson's correlation coefficient r = 0.58; p = 0.0028 two-sided t-test), and to the atmospheric forcing of the North Atlantic Oscillation (Pearson's correlation coefficient r = −0.60, p = 0.003 two-sided t-test), which are both tightly coupled[42]. This suggests that the increase in surface advection in the EAC is likely due to a natural multidecadal oscillation related to the upstream synoptic oceanic circulation and atmospheric forcing, which could, in turn, drive long-term climatic change in the BS[43].

**Increasing NAW surface currents control the spatial distribution of *Ehux*.** The first hypothesis that ocean currents control the summer spatial distribution of *Ehux* in the BS was tested using a Lagrangian experiment (EXP1). We advected virtual particles

(here considered as the inoculum of *Ehux* cells), using observations of surface geostrophic velocities, from their expected overwintering location in March (defined by sea surface temperature >4 °C and distance from coast ≤180 km: see material and methods and Supplementary Note 2.5 and 2.6 for more details on the areal distribution and timing of the inoculum). EXP1 revealed that the spatial distribution of the virtual particles at the end of the advection period matched the extent of *Ehux* blooms (evidenced by satellite-derived Particulate Inorganic Concentration, PIC), with 80% of tracked particles ending up within 50 km of an *Ehux* blooming location (see Supplementary Note 2.7). The spatial distribution of particles in 1998 (beginning of the satellite ocean-color time-series) and 2015 are shown in Fig. 4, while the entire 1998–2016 period is displayed in Supplementary Fig. 14. The year-to-year robustness of the matchup between virtual particles and PIC clearly supports the fact that the NAW surface currents shape the location and extent of *Ehux* blooms in the EAC.

**Increasing NAW surface currents control the poleward expansion of *Ehux*.** It is noteworthy that the particles reached further north and east in the EAC in 2015 than in 1998, in agreement with *Ehux* blooms (Fig. 4). The north-eastward expansion of the *Ehux* bloom, expressed here as the distance reached by the leading-edge (see the Supplementary Method 1.1), increased on average by 325 km during the last 19 years as indicated by ocean-color observations (Fig. 5a), in close agreement with previous estimations (324 km, 40–50°E, 1989–2016[16];) and with the estimates from EXP1 (424 km, Fig. 5b).

The 4 °C surface isotherm (Fig. 4) is considered for *Ehux* as the lowest temperature required for sufficient growth to allow bloom formation (Supplementary Note 2.5). However, the *Ehux* blooms do not seem to follow this isotherm in summer and appear to be

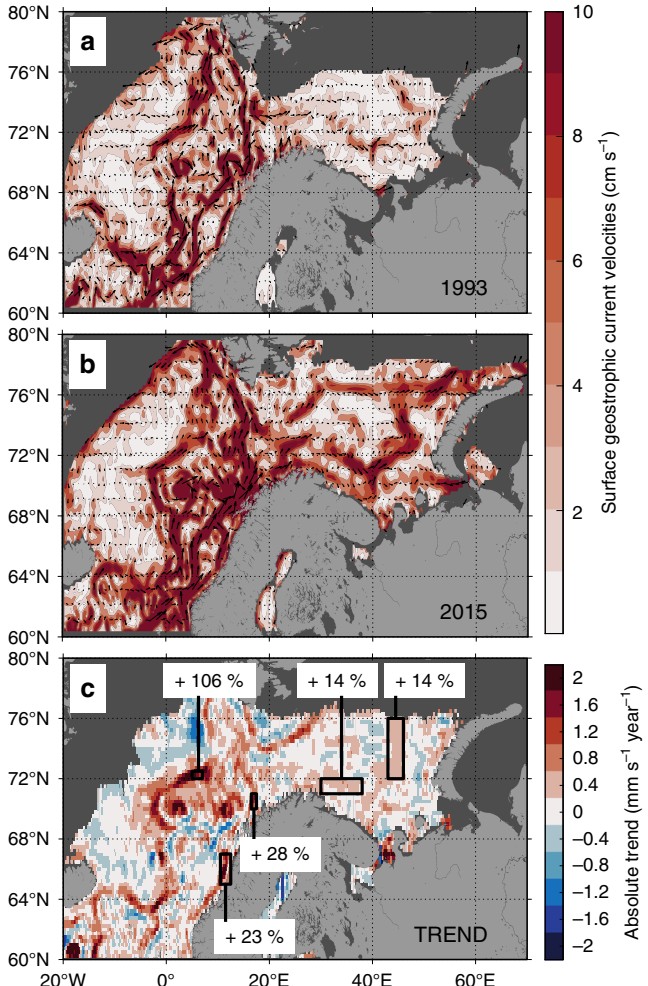

**Fig. 3 Increasing current velocities in the European Arctic Corridor.**
Surface absolute geostrophic velocities during the extremums of the time-series which are, respectively, reached in December 1993 (**a**, minimum) and 2015 (**b**, maximum) with the corresponding absolute linear trend of the entire time-series (all months) over the 1993–2016 period (**c**). Areas covered by sea ice (sea-ice concentration > 15%) or with insufficient data coverage for the trend (<50%) are in dark gray.

constrained by another factor. For example, the north-eastward expansion of the winter 4 °C isotherm, which delimits the areal extent of the inoculum in our Lagrangian method, could contribute to reduce the distance between the *Ehux* winter location and the Arctic domain. To test this hypothesis, we performed two additional Lagrangian experiments to examine both the role of currents and winter temperature on the poleward expansion of *Ehux*. In EXP2 we constantly initialized the virtual particles at the same inoculation region for all years, using a climatological mean temperature field to determine the inoculation region. Hence, EXP2 exclusively reflected the role of currents on the interannual variability of the advected particles. In contrast, in EXP3, the constant inoculation region varied from one year to another, but we used a climatological mean current to advect the particles. In this way, the interannual variability of the advected particles due to interannual variability of the inoculation region was quantified. The interannual PIC leading-edge location was highly correlated with the leading-edge from EXP2 (Pearson's correlation coefficient $r = 0.71$, $p < 0.01$ two-sided $t$-test), suggesting a stronger control of the bloom expansion by currents than by winter temperature (EXP3, Pearson's correlation coefficient $r = 0.43$, $p = 0.06$ two-sided $t$-test). On the decadal scale, currents (EXP2) were found responsible for the 56% (240 km) increase in the long-term leading-edge expansion against 44% (186 km) for winter temperature (EXP3), when compared to the reference EXP1. This significant increasing trend in current velocities (see Supplementary Note 2.8) was also revealed by the greater distance covered by the virtual particles reaching the BS, which increased by 110 km on average since 1993 (EXP2, Supplementary Note 2.9).

Increasing water temperature has previously been assumed to be the main driver of the spatial distribution of *Ehux* bloom in the BS[15,16,44]. Our results demonstrate that the primary driver of the *Ehux* dynamics (i.e., spatial distribution and timing) could be, in fact, stronger surface currents, which in turn intrinsically shape the temperature field and frontal structures. We show that oceanic currents (i.e. their intensity and fluctuations) drive the spatial distribution of the bloom, its interannual variability and more than 50% of the long-term poleward expansion of *Ehux* bloom in the BS. More importantly, from 2006 and onward, the contribution of water temperature to the expansion of *Ehux* bloom becomes negligible (Supplementary Table 3), and its poleward expansion is entirely due to the accelerating currents.

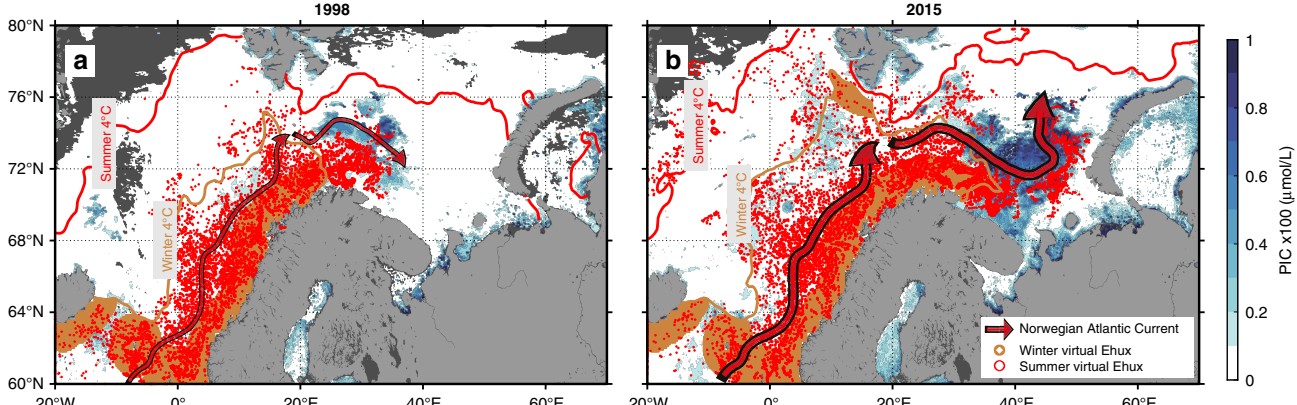

**Fig. 4 Poleward expansion of *Emiliania Huxleyi* (*EHux*) in the European Arctic Corridor.** Comparison between 1998 (**a**) and 2015 (**b**). The initialization (inoculum) of virtual particles in March are illustrated by brown dots. During 6 months, particles drift with the Norwegian Atlantic Current (red arrows) as the ocean seasonally warms as illustrated by the northward expansion of the 4 °C isotherm. In August, the particles end up in positions indicated by the red dots. In the background, remotely sensed PIC indicating coccolithophore biomass in summer (July–August–September) is shown in blue colors. Areas with no data are in dark gray.

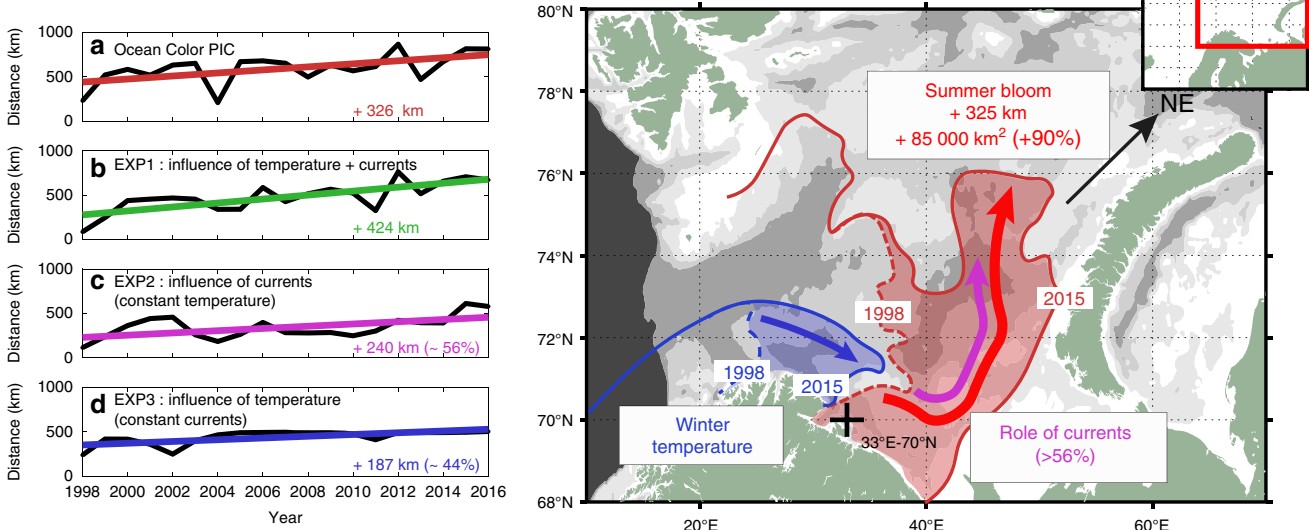

**Fig. 5 Shifting position of the leading-edge *Ehux* bloom distribution.** Shifting position from ocean-color PIC (**a**), and the 3 Lagrangian experiments (**b–d**) for the last 19-years (1998–2016). The comparison between the 1st Lagrangian experiment (EXP1, **b**) with the 2nd (EXP2, **c**) and the 3rd (EXP3, d) aims at estimating the relative contribution of currents (EXP2, constant temperature) vs. temperature (EXP3, constant currents) on the total *Ehux* poleward expansion (EXP1, varying temperature and currents). The right panel is a schematic illustration of the poleward expansion of the *Ehux* with the winter 4 °C isotherm (lowest temperature for a 'regular' *Ehux* growth) in blue and the summer bloom position (northern boundary) in red. The two extreme years 1998 (dashed) and 2015 (solid) are represented. Arrows indicate the contribution from temperature and/or currents keeping the same color code.

**A newly "Atlantified" and retreating marine Arctic ecosystem.** *Ehux* is largely studied given its significant role in marine geochemical cycles[45,46] as illustrated by its sensitivity to ocean acidification[47], its effect on pCO2 and CO2 uptake[48] and its role on carbon export by providing calcite ballast effect[49]. Here, *Ehux* was used as an indicator of Atlantic ecosystems[32,50]. By expanding poleward and doubling its areal extent in the BS (Supplementary Note 2.10), *Ehux* attests to the ongoing "Atlantification" of the Arctic Ocean. Both arctic and Atlantic domains have distinct ecological signatures[51], and the latter is undeniably "invading" the former[4,52]. Advected with the surface currents, *Ehux* will have to survive during the travel (e.g., avoid grazing[53,54], subduction under the polar mixed layer in Fram Strait[55]) until finding more favourable blooming conditions in the BS in summer[13]. The fate of *Ehux* in the BS is therefore of major importance as it determines the potential "seeding effect" of *Ehux* in the Arctic regions. *Ehux* seems to be adapted to the low light, low nutrient, oligotrophic and highly stratified conditions of the NAW in summer[30] such that its expansion, growth and blooming in the Arctic Ocean will be limited at some point by those constraints (bottom-up) but also by the grazing pressure (top-down).

Despite its adaptation to low light conditions, *Ehux* still requires sufficient light levels to sustain the energy-demanding calcification of its coccoliths. Such conditions are met in highly stratified oceans where *Ehux* can accumulate in the surface layer[56]. At high latitudes (>81°N), even with sea surface temperature above 3 or 4 °C, the survival of *Ehux* would require adaptation to rapidly decreasing solar radiations in late summer[57]. *Ehux* fate thus mainly relies on its ability to drift, with the appropriate timing, to highly stratified and temperate areas that allow it to stay in the surface euphotic layer. These conditions[58] would likely be met in the Eurasian interior shelves of the Arctic Ocean[28] (i.e., Kara, Laptev, and Siberian seas) where surface waters are warming and freshening[59]. If the increase in advection along the shelf slope continues in the future, we can expect *Ehux* to become a summer resident of the newly "Atlantified" Eurasian interior shelves, as previously revealed during the last interglacial[50,60].

By driving such a poleward expansion, advective processes could affect the entire marine ecosystems by shifting species distribution[61] and modifying interactions at higher trophic levels[62]. The concomitant decline of silicate concentrations in NAW[63] may also contribute to the success of non-silicifying and small phytoplankton such as *Ehux*[64]. In addition, a change toward temperate pelagic species could have an impact on energy transfer to higher trophic levels[65] including marine mammals and commercial fish stocks[66]. Considering the role of "bio-advection" in ecological models (i.e., trait-based and niche-based approaches) must improve predictions of community shifts[67]. The comparable increase in poleward advection of Pacific waters occurring in the Bering Strait[68] suggests that the shrinking polar domain of the Arctic Ocean may be prone to intrusions of temperate species at a pan-Arctic scale.

## Methods

**Satellite altimetry**. The satellite altimetry product was generated by the Sea Level Thematic Assembly Center (SL-TAC) and distributed by the Copernicus Marine and Environment Monitoring Service (CMEMS, http://marine.copernicus.eu/). This is a daily gridded global product in delayed-time (re-analysis product ID: SEALEVEL_GLO_PHY_L4_REP_OBSERVATIONS_008_047) with a spatial resolution of 1/4° available from 1993 to present time. A 1/4° Mercator projection grid roughly corresponds to a resolution of 27.8 km in latitude × 9.5 km in longitude at 70°N. This product is generated using Sentinel 3 A, Jason 3, HY2A, Saral/AltiKa, Cryosat 2, OSTM/Jason 2, Jason-1, Topex/Poseidon, Envisat, GFO, ERS-1/2. The SL-TAC produced this dataset with a special care for synchronization, homogenization, cross-calibration, correction of large-scale biases, and cross-validation. Before merging missions by optimal interpolation, the SL-TAC filtered the dataset to remove residual noise. The reader is referred to the product documentation for more details. The satellite altimetry product provides Sea Level Anomaly that estimates the variation of the Sea Surface height around a vertical reference (Mean Sea Surface). The product also provides Absolute Dynamic Topography (=Sea Level Anomaly+Mean Dynamic Topography), geostrophic velocity anomalies and absolute geostrophic velocities (derived respectively from Sea Level Anomaly and Absolute Dynamic Topography). The Mean Dynamic Topography employed by the SL-TAC was the MDT-CNES-CLS13 produced by the Collecte Localisation Satellites Space Oceanography division and distributed separately by Aviso (http://www.aviso.altimetry.fr).

SLA has been successfully used over the last 25 years to study the variability of sea level and surface geostrophic circulation at different spatial and temporal scales. However, most of the validation studies focused on the areas below the polar circle (<66°N) such as in the Norwegian Sea where derived geostrophic currents were successfully evaluated against in situ current meters[69]. The use of altimetry in

Arctic regions has been avoided in the past due to (1) persistent sea-ice cover and (2) limitation of some radar altimeters (TOPEX/POSEIDON, Jason-1, OSTM/Jason-2 and Jason-3 missions), which do not acquire measurements above the polar circle. First, sea ice was not an issue for this study because virtual particles were only evolving in the year-round ice-free Atlantic domain. For the trend and EOF calculation, data associated with sea ice were systematically precluded (see below). Second, measurements above 66°N can rely on several other satellites: ERS-1 and -2, Envisat, SARAL/AltiKa, GFO (<72°N), HY-2A, ICESat, Cryosat-2, Sentinel-3. A comprehensive validation of the gridded product of geostrophic velocities was recently provided for ERS-1/2 and Envisat satellites using surface drifters and tide gauges in the whole EAC, with a particular focus on the Barents Sea[70,71]. The authors demonstrated that RMS differences between the drifter (corrected for Ekman currents) and altimetry velocities ranged within 7–15 cm s$^{-1}$, which was comparable to previous estimates at lower latitudes. More importantly, they also concluded that the dataset provided sufficient spatial and temporal coverage to consistently resolve the synoptic and large mesoscale variability. Moreover, the product was recently found appropriate to monitor the long-term northward Atlantic inflow in the Norwegian Sea[40] with even lower RMS differences between drifters and satellites velocities (0.7–8 cm s$^{-1}$).

The dataset used in this study was enhanced from additional satellites (i.e., CryoSat-2 and Sentinel-3A) that offered a better data coverage at high latitudes for recent years (since 2011). Surface currents inferred from altimetry were found to reflect the variability of the deeper layers in the slope current in the Nordic Seas[40]. For this study, the derived surface geostrophic velocities were evaluated again toward five long-term in situ current meters in the southwestern Barents Sea and showed reasonably good Pearson's correlation coefficients (r = 0.43–0.71, Supplementary Note 2.1). In addition, by evaluating the effect of steric and mass related effect on the SLA, we demonstrated that the increasing long-term trend was driven by change in currents (Supplementary Note 2.2).

**Satellite sea surface temperature (SST)**. We used the NOAA 1/4° degree and daily Optimum Interpolation Sea Surface Temperature version 2 (OISSTv2, provided by the NOAA/OAR/ESRL PSD, Boulder, Colorado, USA, https://www.esrl.noaa.gov/psd/data/gridded/data.noaa.oisst.v2.highres.html[72]) to extract the 4 °C isotherms during the month of March between 1998 and 2016. This dataset was constructed by combining satellites observations from the Advanced Very High-Resolution Radiometer (AVHRR) and the Advanced Microwave Scanning Radiometer on the Earth Observing System (AMSR-E) with in situ platforms (ships, buoys) on a regular global spatial grid. Missing observations in SST were optimally interpolated. The dataset was delivered with error fields providing a measure of confidence. This allowed us to build monthly composites maps excluding values with estimated errors higher than 50%, following the same method used for the altimetry dataset. Note that the dataset also provided daily sea-ice concentration maps, which were used as masks for the altimetry dataset. Additional information can be found here: https://www.ncdc.noaa.gov/oisst.

**Satellite particulate inorganic carbon (PIC)**. Ocean colour satellites detect coccolithophore blooms because they produce calcite plates (coccoliths) that shed light and produce 'turquoise' colored waters. Remotely sensed ocean-color offers a 20-year time-series from 1998 to present. In this study, we used the GlobColour database (http://hermes.acri.fr), which provides a continuous dataset of ocean-color satellite data products derived from the merging of five ocean-color sensors MEdium Resolution Imaging Spectrometer (MERIS), Moderate Resolution Imaging Spectroradiometer (MODIS), Sea-viewing Wide Field-of-view Sensor (Sea-WiFS), Visible Infrared Imaging Radiometer Suite (VIIRS) and Ocean and Land Colour Imager (OLCI-A). These sensors measure visible and near-infrared solar radiation reflected back from the ocean surface layer. Such remotely sensed information is only available during the daytime and in the absence of ice and clouds.

Level 3 PIC products were derived from NASA's standard PIC algorithm[73,74]. This algorithm is based on a robust relationship between the light backscattering coefficient and the concentration of coccoliths, calcite plates forming the coccosphere of the coccolithophores[30,75]. Some coccolithophore species, including *Ehux*, produce and release coccoliths into the water during the later stages of a bloom[76–78], creating large patches of highly reflective waters, which can be observed from space[56]. Satellite-derived PIC concentration provides a good proxy for *Ehux* coccolith concentration because *Ehux* dominates the coccolithophore population in the Norwegian Sea[79] and in the Barents Sea[80]. Moreover, satellite PIC products have been successfully validated during *Ehux* blooms in the Barents Sea[13,80,81], as well as in the global ocean[56,74,82].

**The leading-edge detection**. The leading-edge was the main metric used in this study to follow the poleward (i.e., northward and eastward) expansion of the Ehux bloom. The leading-edge was derived as the 95th percentile of the longitude and latitude positions of the virtual particles entering the eastern Barents Sea. See Supplementary Method 1.1 for a more detailed description.

**Empirical Orthogonal Function analysis of sea level and trends in surface geostrophic velocities**. Empirical Orthogonal Function (EOF) and trends analysis were performed between 1993 and 2016, with monthly composite maps of non-seasonal Sea Level Anomalies (SLA) and surface absolute geostrophic velocities. Monthly composite maps were derived excluding pixels with error estimates (percentage of the SLA signal variance) higher than 50% or sea-ice concentrations higher than 15%. The seasonal cycle, known to be highly variable for SLA and the velocity fields, was removed because the analysis focused on the interannual and decadal time scale. The maximum amplitude in SLA generally occurs in fall[83], whereas the velocity fields are maximum during the winter[84]. More details on the EOF analysis are available in Supplementary Note 2.3. Trends in surface absolute geostrophic velocities were only derived for pixels associated with more than 50% of data points.

**Lagrangian calculation of particles trajectories**. The Lagrangian experiments needed daily temporal resolution and directly employed daily absolute surface geostrophic velocities delivered by CMEMS. Three different Lagrangian simulations were used to address specific objectives. In each simulation, particles were advected with a 4$^{th}$ order Runge-Kutta integration scheme, with a time step of 3h. The simulations showed good robustness to the addition of noise, which was representative of the velocity fields uncertainties (Supplementary Note 2.4). The different simulations are described below.

(1) Variable temperature and currents. The first simulation (EXP1) explored the influence of seasonal current variability on Ehux distribution from 1998 to 2016. On the 1st of March of each year, we selected a patch of particles representing the theoretical inoculum area of *Ehux* (i.e., SST ≥ 4 °C and distance from coast < 180 km, see Supplementary Note 2.5). These particles were advected, using satellite-derived velocities, over the entire growing season, when sufficient light availability is expected to allow phytoplankton growth (i.e., from the March 1st to September 1st). The choice of March 1st as the starting date is discussed in Supplementary Note 2.6.

(2) Constant temperature and variable currents. The second simulation (EXP2) investigated the long-term increase in particle advection in the BS (1993–2016). In this simulation, instead of having a distinct 'inoculum patch' for each year, we inoculated the same patch every year. This patch was built using the March SST climatology (1993–2016) and the same criterion than EXP1 (SST ≥ 4 °C and distance from coast < 180 km). The particles were then also advected during the entire growing season (i.e., from March 1st to September 1st) as in EXP.1.

(3) Variable temperature and constant currents. The third simulation (EXP3) aimed at assessing the role of temperature in the long-term poleward expansion of *Ehux*. EXP3 is similar to EXP1 with the exception that the particles are advected each year on the same climatological velocity field (varying 'inoculum patch').

**Statistical analysis**. Trends were estimated using linear least-squares regression. Correlations were calculated using Pearson's coefficient correlation r. Two-sided t-tests with N–2 degrees of freedom were used to find significant levels (p-values) for trends and correlations. The significance level for this study was considered as 95% ($p < 0.05$).

**Reporting summary**. Further information on research design is available in the Nature Research Reporting Summary linked to this article.

## Data availability

All the data used in this research are freely available to the public and may be downloaded through the links detailed in the Methods section. Analysis outputs data for the main figures are made available at https://gitlab.com/loloziel/oziel_et_al_nc_2020.

## Code availability

The code for Lagrangian experiments is based on Lamta 0.2 Copyright (C) 2007/12/5 Francesco d'Ovidio (francesco.dovidio@locean-ipsl.upmc.fr). This is a free Octave software under the terms of the GNU General Public License (http://www.gnu.org/licenses/) and can be distributed and modified. Outputs were analyzed using ©Matlab custom scripts that are accessible at https://gitlab.com/loloziel/oziel_et_al_nc_2020.

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

## Acknowledgements

L.O. was funded by an NSERC/CRSNG "Visiting Fellowship" in Canadian laboratories in the framework of the DFO-International Governance Strategy (IGS). L.O. was also supported by the joint international laboratory Takuvik (ULaval/CNRS), the Canada Excellence Research Chair in remote sensing of Canada's new Arctic frontier. This research was supported by the Sentinel North program of Université Laval, made possible, in part, thanks to funding from the Canada First Research Excellence Fund. M.A. was supported by a European Union's Horizon 2020 Marie Sklodowska-Curie grant (no. 746748). The authors would like to thank the international agencies and programs from the Copernicus Marine Environment Monitoring Service (CMEMS), the National Oceanic and Atmospheric Administration (NOAA), and the ACRI-ST company for freely providing access to their satellite altimetry, Sea Surface Temperature and Ocean-color data. This work heavily relies on the following toolbox for mapping: Pawlowicz, R., 2018. "M_Map: A mapping package for MATLAB", version 1.4j, [Computer software], available online at www.eoas.ubc.ca/~rich/map.html as well as environmental toolbox developed by Chad Greene (2019): The Climate Data Toolbox for Matlab (https://www.github.com/chadagreene/CDT), GitHub. Retrieved March 11, 2019. The authors thank Francesco D'Ovidio for the Lagrangian tools, Martine Lizotte for language support and Marie-Isabelle Pujol for her advices on satellite velocity uncertainties.

## Author contributions

L.O. conceptualized the study, conducted data analysis, and wrote the manuscript. A.B. performed the Lagrangian experiments. M.A. helped in the writing and in the conceptualization of the manuscript. P.M. and R.B.I. helped in the analysis of the steric contribution to the SLA. R.I. provided current meters data and helped in the validation analysis. A.R. derived the virtual particles match-ups. All authors (L.O., A.B., M.A., P.M., A.R., J-B.S., R.B.I, E.D., and M.B.) contributed to the ideas and edited the manuscript.

## Competing interests

The authors declare no competing interests.
