## [Peer Review File · Nature Communications]

Reviewers' comments:

Reviewer #1 (Remarks to the Author):

The subject of this manuscript is very timely. No other place on the globe is global warming more prominent than in the Arctic, particularly around the Svalbard archipelago. And the atlantification is probably the most distinguished sign of that warming. To use remote sensing in manner exemplified by the authors is praise worthy. Despite of that I am not in favour to support publication, at least not in its present form.

Before I continue I have to note that I am not a specialist in remote sensing and oceanographic modelling and that technical details in this paper cannot be evaluated by me. I keep to the advection of phytoplankton and present and future phytoplankton blooms.

I was in doubt if I understand well what the authors mean by advection? It is the spread of particles into motherly directions, but is this caused by increased volume transport of Atlantic Water or increased velocities or increased velocities at the surface? In case they consider it an increased volume transport, why do they consider so lightly the work of Igor Polyakov and colleagues? He is the authority to refer to. That this is not done should, in my view, not be accepted.

And while I focus upon that I should add that salient literature centred around the work of Griet Neukermans and others is not adequately cited. Several published papers on modelling and advection are not cited, among those works of Dag Slagstad, Maria Vernet and Paul Wassmann. That is unfair when it comes to the authors and does not pay justice to the results of the manuscript.

Please remove anthropocentric words such as dramatic, drastic etc throughout the manuscript.

There are incongruences when it comes to citations. It looks like that the authors have not paid sufficient attention to this before submitting. Was it done hastily?

I wonder how to interpret this: ADTEOF1,2, where EOF1,2 is raised?

How do you define the term thermal barrier?

"Ehux is largely studied given its significant role in marine biogeochemical cycles". That is not true in the manner you write it. They may play a role in the geochemical cycles, but there is so to say no biomass in Ehux blooms. Nor do they have increased primary production. Ehux is studied because

their significant in the geological record, as tracer for water masses and, last not least, because they are visible from space.

How do I understand "Atlantic-like environmental conditions and ecosystem"?

I have never heard earlier that *Ehux* is "Thriving with currents" and that because of that "*Ehux* either experiences favourable growth conditions to bloom at the surface such as in the BS"?

While discussing the blooms of *Ehux* that authors ignore grazing by the microbial loop or dissolution. This is an essential process in any bloom. The interpretation is only "bottom-up" and thus unacceptably simplistic. In particular when the authors then point at higher trophic levels as sign of climate change. The connection between blooms, haverstable production and higher trophic levels has to be fully considered when *Ehux* is characterised by a significant role in biogeochemical cycling.

The authors mentioning of the solar cycle and light in the marginal ice zone is unacceptable primitive. What is a short solar cycle when the ecosystems north of 81°N have almost continues light for 6 months?

Pease revise your terminology when it comes to interior shelves (see Carmack and Wassmann 2006).

Bioadvection of ecological models: there far better ones than those that were omitted cited.

Reviewer #2 (Remarks to the Author):

In their manuscript, Oziel and coauthors test a very interesting hypothesis that the expansion of *Ehuxleyi* in the Barents Sea is resulting from a recent increase of currents flowing from the Atlantic. The expansion of this species has been attributed to temperature or CO₂ concentration increase, but the transport hypothesis as never been discussed yet. It is natural to think that if the species does not stay in the Arctic during the winter because of darkness, they would have to migrate every spring and summer to those latitudes, and then because *Ehux* is part of the plankton (meaning that it cannot move against a current) it has to be transported. It is indeed difficult to imagine *Ehux* surviving the winter night at the latitude of the Barents Sea, and the self migration on hundreds of kilometers for an organism of 6µm in diameter is impossible. The work is essentially based on satellite observations compared with Lagrangian simulations of currents. They show that the Atlantic current speed increase recently and therefore penetrate more the Barrents Sea. The coccolithophore bloom migration by flow transportation is the easiest interpretation. The experiments indicate that *E.huxleyi* can be transported to those latitudes by the currents and

therefore could explain the polar expansion of *E. huxleyi*. The Lagrangian simulations of the current in the different year, as presented in this work, surprisingly track the decadal expansion of the bloom. The demonstration is quite elegant and the text is well written. This is remarkable and therefore I recommend its publication in Nature Climate Change.

I have some comments and questions :

1- the term 'inoculation point' is not well adapted to this context, because it may be understood as a migration starts from just one point, which is not the case. I would like to read a clear description of the surface of origin of the bloom. Unfortunately, in Figure 9 in supplementary material, the blue dots do not appear on the map. 2- This figure should be redrawn (showing blue dots).

2-A question arrive from Figure 10 in the suppl. Mat. : Why the simulated surface of the bloom (in red) is always smaller that the observed one (in grey). Could it be that the 4°C winter isotherm does not represent the best estimate of the winter edge expansion and instead it should be something colder like 3°C which should represent a larger surface of origin ? In any case, this figure is fascinating and should be placed in the main article.in replacement of Figure 4, or even 3 which is difficult to read.

3-The polar expansion of *Ehux* bloom has been first described by Winter et al., in 2014. It is surprising that this work is not mentioned in the main text. The only mention of it is in the supplementary material but with a wrong author ranking (Brown is placed as a first author) and with a wrong date (2013 instead of 2014). A description of this study should be discussed earlier in the paper, because it is well in line with this work and again is the seminal paper on that subject.

4- In general the reference list is poorly written. Many errors are found (see details below)

5-The term 'niche' is not appropriated because here it relates only to temperature; a niche being something more complex involving light, nutrients among many other parameters. Terms like 'temperature limit', 'lowest temperature requirement for bloom development', may be more adapted. The concept of niche in ecology is much broader than what is meant here (Also in the Supplementary material section, the description of the niche in term of temperature is too limitative and provide a very narrow temperature range - In the ocean, *E. huxleyi* has ben found in temperature ranging from -1°C to 30°C. It blooms in the Atlantic at around 14-16°C. In culture is best development is at around 20°C.

6- What would be the conditions that would make *Ehux* bloom in the arctic. Would it be light with only a temperature limit (>4°C)? That is a bit short view. Also if a cell stays in the same body of water and divide to reach bloom, why this body of water does not become sooner nutrient depleted ?Here the blooming cells would stay from March to August in the same water. In the rest of the ocean, blooms occur for few weeks and collapse. How does the multiplication by cell division is taken in consideration in this work ? Is there any type of mixing of the Atlantic water mass with local waters, that could help bringing new nutrients to fuel the bloom development. I do not see any mention of that.

7 The end of the discussion is quite speculative and also may appear in contradiction with the main findings : It is said that *E. huxleyi* could penetrate permanently to East in the southern shelves of the Arctic Ocean : the problem that is foreseen is that 1) the winter darkness, which prevents *E.huxleyi* to survive the winter (this is the first postulate of the manuscript), will not change in the future, 2) if this future yearly colonization by *E.huxleyi* of the Arctic Ocean is due to ocean warming, then one may wonder why it did not do that in the past since *E. huxleyi* is a eurythermal species.

Minor remarks :

Line 49 : NAW 'carry most of the heat' : imprecise and not clear

Line 81 : deepest should be replaced by something like 'farthest'

Line 114: +2mm s-1 yr-1 does not seem very much (like 172 m /day or 15 km in 3 months) : can you comment on this ?

Line 156 : why Ehux blooms do not follow the surface thermal barrier when later it is said that the thermal barrier results from Atlantic water advection.

Line 159 : 'could play a critical role by' could be replaced by 'help'

Line 163 & 164 : 'at the same location' seems in contradiction with determining the inoculation region. A clarification is needed.

Line 177: Winter et al., 2014 should be quoted... and also (as I mentioned earlier) it should be discussed in the introduction since it is the first mention of Ehux Polar Expansion.

Line 202 : the reference is not written well in the reference list (see below).

Line 204 : 'short solar cycle' - No - short night or long days but the solar cycle is not shortened : 24 hours in most of the cases or 1 year at the north pole .

Line 205 : 'warm' -> relatively warm : Arctic is not tropical :-); 'Rest' - not appropriated.

The reference list is poorly written :

For exemple :

Ref 5. Beszcynska-mo for Beszcynska-Moëller and Marine Science written twice, and no pages.

Ref 13 <<i&.... In the title

Ref. 28 BIRKENES & E ??? Who is E ?

Ref 44 the beginning of the Abstract is written instead of the title and journal.

Ref 26 of Supplements : Author are not in order and a wrong year (should be 2014).

Reviewer #3 (Remarks to the Author):

Review, Oziel et al 2019: Faster Atlantic currents drive poleward expansion of temperate marine species in the Arctic Ocean.

This paper bases its conclusions on two fundamentals:

A velocity field estimated from satellite altimetry for the time period 1993 to 2016

A distribution of *Emiliana Huxleyi* estimated from satellite observations of Particulate Inorganic Concentration for the same time period.

There is a change in the leading edge of the PIC in the Barents Sea during that period: northeastward. They use the (changing) velocity field to trace lagrangian particles and conclude that the moving leading edge can be explained primarily by an increase in velocity of the Atlantic Water and not so much by the increase in temperature of the Atlantic Water, as was previously assumed.

The main problem I have with this manuscript is that the authors do not investigate the velocity field as thoroughly as they ought to. Most investigations of the heat budgets of the Barents Sea and the Arctic Ocean rests on current meter measurements at the Barents Sea opening, of which there are now nearly 20 years of data. And even before 1997, there were several published estimates of that transport. I have tried to collect them in the list below.

Timofeyev, 1963: 2 Sv Atlantic Water eastward, 0.9 Sv westward return flow, net east: 1.1Sv

Blindheim, 1989: 3.1 Sv Atlantic Water eastward, 1.2 Sv return westward flow, net east: 1.9 Sv.

Mauritzen, 1996: net east: 1.6 Sv

Skagseth, 2008: 2.0 Sv Atlantic Water eastward (time period 2003-2007).

Skagseth et al. 2008 (Volume and heat transports to the Arctic Ocean via the Norwegian and Barents Seas.): 1.8 Sv net east, time period 2000-2006, with a positive trend from 1 Sv in 2006 to 2 Sv in 2006.

Smedsrud et al, 2010: 2 Sv Atlantic water eastward, 1.2 Sv Norwegian Coastal Current Water eastward, 1.2 Sv westward return flow, net east: 2 Sv. (time period 1997-2007)

Årthun et al 2012: 2.3 Sv eastward, time period 2000-2009, with a positive trend.

As one can tell from these numbers, it is possible to argue that the volume flux has increased during the 2000s [but the changes are not much larger than the expected error bars of the measurements]. Nevertheless, the authors have not made these comparisons. The two estimates showing positive trends (Årthun et al and Skagseth et al) display completely different time evolution, and neither one resembles this manuscript's figure 2B.

Without a thorough discussion of their velocity findings, in the context of previous publications and the observations made in the area, this reviewer has problems accepting the velocity results upon which the paper is based.

But assuming that this manuscript's velocity fields are representing reality, the next "problem" is that the advection of EHUX is over-explained by the changes in velocity and temperature (see figure 5). Including both effects, the particles are advected 100 km further than they should have been (which is 326 km). Including only temperature effects, the particles are advected 140 km short of the target. I.e. the temperature explanation gets us 187 km towards the target, and one only needs another 140 km to reach the target. Therefore it is misleading to state (as is done in the abstract) that "we further demonstrate that bio-advection, rather than water temperature as previously assumed, is the major mechanism responsible for the recent intrusion....".

Let me be clear: I find the paper intriguing, and possibly publishable. But the authors need, in my opinion, to 1) make a better case for their velocity field, and 2) to reformulate their major findings to align with the actual results.

Dear Editor, dear reviewers,

We would like to thank you for the advices, comments and constructive critics. They allowed us, we believe, to improve significantly the quality of the manuscript. Please, find below our point-by-point answers. You can also follow modifications on the tracked-change manuscripts (main text and Supplementary Information).

Best regards,

Laurent Oziel, on behalf of all co-authors.

Reviewer #1 (Remarks to the Author):

The subject of this manuscript is very timely. No other place on the globe is global warming more prominent than in the Arctic, particularly around the Svalbard archipelago. And the atlantification is probably the most distinguished sign of that warming. To use remote sensing in manner exemplified by the authors is praise worthy. Despite of that I am not in favour to support publication, at least not in its present form.

Dear Reviewer,

We would like to thank you for your comments which have helped us improving the quality of the manuscript. We hope the numerous modifications will answer your concerns.

Before I continue I have to note that I am not a specialist in remote sensing and oceanographic modelling and that technical details in this paper cannot be evaluated by me. I keep to the advection of phytoplankton and present and future phytoplankton blooms.

I was in doubt if I understand well what the authors mean by advection? It is the spread of particles into motherly directions, but is this caused by increased volume transport of Atlantic Water or increased velocities or increased velocities at the surface? In case they consider it an increased volume transport, why do they consider so lightly the work of Igor Polyakov and colleagues? He is the authority to refer to. That this is not done should, in my view, not be accepted.

This study mainly evaluated the inferred surface geostrophic velocities (i.e. representative of the mixed layer, around 50 meters) from satellite altimetry sea level (not modelled). However, we realize that this was not properly described in the previous version of the manuscript, and we have now systematically and carefully re-frame the manuscript to the surface ocean currents only to avoid further confusion. We use the term advection (or bio-advection) as the spread of particles into northerly (or easterly) directions, and this is caused by increased velocities in the surface layer/upper 50 m water layers. When we use the term advection, it is only for this uppermost layer. Thus, this cannot be compared to published

transport estimates of Atlantic Water covering the full water column (for explanation, see Supplementary Material 2.1). More details on this issue is given under the answer of reviewer 3.

Concerning the work by Polyakov and colleagues such as the one published in 2017 in Science (<https://science.sciencemag.org/content/356/6335/285>), we agree that such a work deserves to be earlier cited in our article (it was already cited in the first paragraph following the abstract: “The NAW inflow largely controls the physical and sea-ice conditions of the region^{9,11–13}”). This is now done in the abstract as soon as the term “Atlantification” is introduced.

And while I focus upon that I should add that salient literature centred around the work of Griet Neukermans and others is not adequately cited. Several published papers on modelling and advection are not cited, among those works of Dag Slagstad, Maria Vernet and Paul Wassmann. That is unfair when it comes to the authors and does not pay justice to the results of the manuscript.

We have done our best to correct this issue. We added Neukermans et al directly in the introductory paragraph (while it was cited just after in the first paragraph after the abstract) when talking about the poleward expansion:

“ [...] **recent poleward intrusions of southern species like Ehux** ^{* Neukermans et al. 2018} “

We did not find any other paper from Griet Neukermans that could help in the contextualization or the discussion of this paper (either about EHux, “Atlantification” or bio-advection).

It is true that in the first sentence of the 3rd paragraph we jumped directly from phytoplankton to higher trophic levels without quoting anything for zooplankton. We now added two references (Gluchowska et al 2017, <https://doi.org/10.1093/icesjms/fsx033>; and Edvardsen et al 2003). We selected these two works because the former is the most recent and complete work on the study area with also the longest time series. The latter is the study that tackles the zooplankton bio-advection the most directly with current meters.

We added Wassmann et al. 2015 to Hunt et al. 2016. Also we found it relevant to quote previous work about “bio-advection” such as Popova et al. (2013, zooplankton advection with models), Carmack and Wassmann (2006, a pan-arctic synthesis on physical-biological coupling that discuss theoretically the advection), Basedow et al. (2018, evaluation of the seasonal zooplankton transport through the Fram Strait), Slagstad et al. (2015, evaluation with SINMOD model of future scenarios for the productivity of the arctic, advection is discussed based on hydrographic data but the advection, i.e. the ocean currents, are not investigated).

We also added the very recent works:

Wassmann et al. 2019: Advection of Mesozooplankton Into the Northern Svalbard Shelf Region

Vernet et al. 2019: Influence of Phytoplankton Advection on the Productivity Along the Atlantic Water Inflow to the Arctic Ocean

As a general comment, we paid a particular attention to improve contextualization with the existing literature about the advection itself, the AW inflow and the bio-advection.

Please remove anthropocentric words such as dramatic, drastic etc throughout the manuscript.

We changed in the abstract “ [...] Our findings confirm the biological and physical “Atlantification” of the Arctic Ocean⁹ with potential alterations of the Arctic marine food web and biogeochemical cycles. “

In the 4th line of the second paragraph, we changed “drastic changes” by “substantial changes”.

There are incongruences when it comes to citations. It looks like that the authors have not paid sufficient attention to this before submitting. Was it done hastily?

We apologize for this negligence. Actually, we faced issues with our bibliography software (Mendeley). To avoid this problem to happen again, we have now turned on a fully manual (and corrected) bibliography so it does not happen anymore.

I wonder how to interpret this: ADTEOF1,2, where EOF1,2 is raised?

$ADT_{EOF1,2}$ (noted in subscript to avoid confusion with citations) is the re-construction on the ADT signal from the EOF analysis. By re-creating ADT from the two main first modes of variability of the EOF following the method described in the former manuscript:

“To re-compose ADT maps corresponding to the positive and the negative phases of the EOF, we considered the extrema of the Principal Components (PC). For instance, the ADT_{EOF} at a time = t and position (x_0, y_0) is derived as: $ADT_{EOF}(t, x_0, y_0) = MDT + EOF(x_0, y_0) \times PC(t)$, where MDT is the mean dynamic topography, EOF is the spatial structure of each mode, and PC is the time-series associated to each mode. The obtained composite ADT fields were used to derive the associated composite surface absolute geostrophic velocities corresponding to the EOF”

We show that the variations in the ADT fields can be attributed to the dominant inter-annual modes of variability (associated with the Atlantic water inflow) and their effects on surface geostrophic velocities. To check that this EOF analysis properly decomposed the ADT variability, we compare with the ADT_{EOF} (first 2 modes of variability, ~80 % of the total variance) with the total ADT (full signal).

We agree that our nomenclature may confuse readers, so we decided to replace ADT_{EOF} by the ADT itself in the main text (i.e. decomposed vs. full signal). We moved the comparison

between the ADT_{EOF} and the ADT in supplementary results. We clarified this in the second paragraph of the section “Climatic modes and increasing NAW current velocities” by framing the result on the Atlantic inflow:

“To reveal the impact of changing sea level on surface velocity fields, absolute surface geostrophic velocities, derived from Absolute Dynamic Topography (ADT) fields, are shown in Fig. 3ab during the extreme negative (i.e. 1993) and positive (i.e. 2015) phases of the EOF. These two contrasting years illustrate the ongoing strengthening of the ocean surface circulation in the EAC associated with the Atlantic inflow. Surface geostrophic current linear trends derived from ADT showed that surface currents changed in most of the Atlantic pathway (Fig. 3c).”

Note that you can track the changes on the *.doc manuscript version 2.

How do you define the term thermal barrier?

Here, we used thermal barrier in the sense of what we now define as the “the lowest temperature required for their growth”. By thermal barrier, we want to say that below a certain temperature (here for example 4 deg C), EHux may survive but is unlikely to grow, therefore preventing Ehux to develop beyond the 4degC isotherm. More information about the EHux minimum temperature for EHux growth can be found in supplementary results 2.5 (The Ehux ecological temperature niche and delineation of the inoculum area).

To clarify, we changed to:

“The 4°C surface isotherm (Fig. 4) is considered as a thermal barrier for EHux because it represents the lowest temperature required for their growth (Supplementary Result 2.5). However, the Ehux blooms do not seem to follow this thermal barrier in summer and appear to be constrained by another factor. For example, the north-eastward expansion of the winter 4°C isotherm, which delimits the inoculum area in our Lagrangian method, could contribute to reduce the distance between the Ehux winter location and the Arctic domain.”

“Ehux is largely studied given its significant role in marine biogeochemical cycles”. That is not true in the manner you write it. They may play a role in the geochemical cycles, but there is so to say no biomass in Ehux blooms. Nor do they have increased primary production. Ehux is studied because their significant in the geological record, as tracer for water masses and, last not least, because they are visible from space.

We would like to argue that coccolithophore have an important role in biogeochemical cycles for many reasons. For example, for their role in the carbon cycle via their calcite shells as been well documented. In the last Nature Geoscience article from Riebesell et al. (2017, DOI: 10.1038/NGEO2854) for example, they listed:

*“ its [coccolithophore] **crucial role in ocean biogeochemistry is well documented.** Coccolithophores are responsible for about half of pelagic calcium carbonate production 4 , which increases Earth’s albedo 5 , decreases oceanic CO uptake through the reduction of*

surface layer alkalinity 6 , and enhances carbon flux to depth by providing calcite ballast to accelerate sinking of organic particles 7,8 .”

In Neukermans et al. 2018, again :

“E. huxleyi plays a crucial role in ocean biogeochemistry and various climate change feedback mechanisms. For example, E. huxleyi is known to alter the Earth’s radiation budget (Tyrrell, Holligan, & Mobley, 1999), decrease oceanic CO₂ uptake through the reduction of surface layer alkalinity (Frankignoulle, Canon, & Gattuso, 1994) and may enhance carbon flux to depth by providing calcite ballast to sink organic particles (Riebesell et al., 2016).”

We however agree to some extent that their contribution to PP is minor and that the export of EHux carbon export is still debated. For those reasons, we agree to change biogeochemical by geochemical. The paragraph becomes:

“Ehux is largely studied given its significant role in marine geochemical cycles^{44,45} as illustrated by its sensitivity to ocean acidification⁴⁶, its effect on pCO₂ and CO₂ uptake⁴⁷ and its role on carbon export by providing calcite ballast effect⁴⁸. Here, Ehux was used as an indicator of Atlantic Waters^{32,49}. By expanding poleward and doubling its areal extent in the BS (Supplementary Result 2.10), Ehux attests to the ongoing “Atlantification” of the Arctic Ocean. Arctic and Atlantic domains both have distinct ecological signatures⁵⁰, and the latter is undeniably “invading” the former^{10,51} ..”

How do I understand "Atlantic-like environmental conditions and ecosystem"?

We agree this a complicated way to simply say Atlantic Waters. We thus replaced by “Atlantic Water”. See modification above and in the tracked-change manuscript.

I have never heard earlier that Ehux is "Thriving with currents" and that because of that "Ehux either experiences favourable growth conditions to bloom at the surface such as in the BS"?

Right, the word “thriving” is not be the appropriate terminology. All phytoplankton are drifting with currents since they do not have the ability of motion. We changed to “Advection with currents, ...” or “drifting with currents” in some other occasions. We did not mean that "Ehux either experiences favourable growth conditions to bloom at the surface such as in the BS" because of the advection. We meant that while they drift with currents, and that along their trajectories, they encounter changing environmental conditions and grazing pressure. And if they bloom at surface in the BS, it means that somehow, they have found favorable conditions (low grazing, stratified water column and oligotrophic conditions for example). But it is true that this formulation does let any room for secondary production consideration. We have entirely re-written the paragraph to better consider grazing:

“Advection with the surface currents, Ehux will have to survive during the travel (e.g. avoid grazing⁵², subduction under the polar mixed layer in Fram Strait⁵³...) until finding more favourable blooming conditions in the BS in summer¹⁸. The fate of Ehux in the BS is therefore of major importance as it determines the potential “seeding effect” of Ehux in the Arctic regions. Ehux seems to be adapted to the low light, low nutrient, oligotrophic and highly stratified conditions of the NAW in summer⁵⁴ such that its expansion, growth and blooming in the Arctic Ocean will be limited at some point by those constraints (bottom-up) but also by the grazing pressure (top-down).”

In this way, the blooming of Ehux can be attributed to both top-down and/or bottom-up controls. We also added the grazing pressure to the list of constraints at the end of the paragraph with references to Olli et al. 2007 and to the text book by Valiela, I. (Marine ecological processes, Springer, 1995). The authors indeed suggested the advection of expatriate zooplankton may consume most of the algal production. We hope that those changes will answer your concerns.

While discussing the blooms of Ehux that authors ignore grazing by the microbial loop or dissolution. This is an essential process in any bloom. The interpretation is only "bottom-up" and thus unacceptably simplistic. In particular when the authors then point at higher trophic levels as sign of climate change. The connection between blooms, haverstable production and higher trophic levels has to be fully considered when Ehux is characterised by a significant role in biogeochemical cycling.

The reviewer is right. Because this manuscript does not investigate secondary production, we first made the choice to avoid discussing the grazing pressure from zooplankton. But the accumulation of the biomass (advected or produced locally) is always constrained by losses. Phytoplankton accumulates as soon as gains overcomes losses. In other words, if a bloom occurs (here a biomass accumulation) it is that the zooplankton is not able to “graze enough” and a mismatch between primary and secondary production appears. Biomass produced elsewhere may be carried by currents to a new location where it may prey be consumed by (top-down) other organisms. At mesoscale, biomass may be concentrated by physical structures (e.g. eddies, convergent fronts) to increase concentration locally and thus their availability to predators.

This concern has already been answered in the previous comment. We hope that this new version answers your concern about the lack of top-down interpretation.

The authors mentioning of the solar cycle and light in the marginal ice zone is unacceptable primitive. What is a short solar cycle when the ecosystems north of 81°N have almost continues light for 6 months?

We want to thank the reviewer for raising this issue which was not properly written in the first version. What we meant is that Ehux bloom in late summer (end of August/ early September) and at that time, the solar radiation is quickly decreasing due to the low solar angle at high latitudes compared to lower latitudes (see illustration below). Thus, at this time of the year, Ehux is short in time to grow under favorable light conditions and advection

further in the poleward direction may not allow them to have enough light to keep on growing. We reformulated as:

“At high latitudes (>81°N), even with sea surface temperature above 3 or 4 °C, the survival of Ehux will require adaptation to rapidly decreasing solar radiations in late summer⁵⁶”

Citation: Pidwirny, M. (2006). "Earth-Sun Relationships and Insolation". *Fundamentals of Physical Geography, 2nd Edition*. Date Viewed. <http://www.physicalgeography.net/fundamentals/6i.html>

Please revise your terminology when it comes to interior shelves (see Carmack and Wassmann 2006).

The reviewer is right. In Carmack and Wassmann (2006), the terminology used is “Interior shelves”, or more specifically “Eurasian interior shelves” in our case. They further detail and classify the Kara, Laptev and Siberian Seas that we are talking about as “Wide interior shelves”. We decided to stick with “Eurasian interior shelves” everywhere in the text.

We changed to:

“ These conditions⁵⁸ would likely be met in the Eurasian interior shelves of the Arctic Ocean³⁰ (i.e. Kara, Laptev and Siberian seas) where surface waters are warming and freshening⁵⁹.”

Reference 30. being Carmack and Wassmann (2006).

Reference 59. Williams and Carmack (2015): <http://dx.doi.org/10.1016/j.poccean.2015.07.008>
0079-6611/Crown

Bioadvection of ecological models: there far better ones than those that were omitted cited.

We are not sure we understand this sentence. If you mean that there are articles about bio-advection that were not cited here, yes, we totally agree. Due to your earlier comments, we have added several studies using bio-geochemical models to help contextualize (e.g. Popova et al., 2013, Slagstad., 2015, Wassmann et al., 2015, 2019). Please, see our answer above for more details.

page 2 line 51: But does this imply that the volume has increased? We know that the temperature has increased. If you see increased velocities than this has to be a surface phenomenon?

This is correct, and should hopefully be clear with the conducted revisions.

page 2 line 54: I see that you do not cite Increased intrusion of warming Atlantic water leads to rapid expansion of temperate phytoplankton in the Arctic G Neukermans, L Oziel, M Babin Global change biology 24 (6), 2545-2553

Why is that? Is previous work that your investigation is based upon not cited? I consider this omission critical. Are the authors really fair?

The article (Neukermans et al. 2018a) was cited just after in the second paragraph but the reviewer is right it may be inappropriately late. The correction has been done as already described above. The article is now cited in the abstract when introducing poleward expansion of Ehux :

“water temperature as previously assumed⁷, is the major mechanism responsible for the recent poleward intrusions of southern species like Ehux⁸”

8. Neukermans et al. 2018

page 2 line 58: What is the drama?

The concern has already been answered. We removed all ‘anthropocentric’ qualifications such as ‘dramatic’.

page 2 line 63: Drastic ?

Same answer here than previous comment. We replaced it by “substantial”.

We deleted as well the “drastic” in the first sentence of the last paragraph on the last section: “By driving such a ~~drastic~~ poleward expansion, advective processes”

page 2 line 68: Citation in text ?

We apologize about this. This has been corrected.

page 2 line 71 : What about citing the careful evaluation of the role of advection in the Arctic Ocean by Wassmann et al. 2015?

Yes, this is a mistake from our part, we apologize. The paper actually quoted when saying bio-advection is an “essential mechanism” is Wassmann et al. 2015 and not Hunt et al. 2016. We corrected and now refer to both articles.

That may be right, but it is it? For zooplankton, see various papers citec in Wassmann et al. 2015.

We do in principle agree, but the letter format of this article does not allow us to be completely exhaustive. The number of words and citations are limited. It is true that the advection mechanism has been more assessed for zooplankton than for phytoplankton. We have tried to re-frame with the existing (most recent) literature about bio-advection of zooplankton. We changed to:

“Although the actual role of advection has already been identified as a potential driver altering zooplankton dynamics^{26–28}, it has never been assessed quantitatively from observations for phytoplankton in the EAC, or more generally in the Arctic Ocean^{6,25,29–31} “

26. Basedow et al. (2018) is about seasonal advection of zooplankton in the WSC from observations
27. Wassmann et al. (2019) underlined the importance of the advection of mesozooplankton for potentially feeding higher trophic levels.
28. Slagstad et al. (2015) Sinmod model simulation about the future primary and secondary production in the Arctic Ocean (no quantitative estimation of the advection)
29. Popova et al. (2013) deals with pan-arctic model and nutrient advection.
30. Carmack and Wassmann (2006) “Food webs and physical-biological coupling on pan-Arctic shelves: Unifying concepts and comprehensive perspectives”
31. Vernet et al. 2019 (accepted manuscript, in press): The authors estimated from model simulation that the advected phytoplankton biomass overcomes the biomass produced locally in the EAC.
6. Hunt, G. L. *et al.* Advection in polar and sub-polar environments: Impacts on high latitude marine ecosystems. *Prog. Oceanogr.* **149**, 40–81 (2016).

This list of articles is not exhaustive. But to our knowledge, this is a good sample of the most recent studies about bio-advection and are synthesizing a state-of-the-art of bio-advection processes for phytoplankton and zooplankton in the Arctic Ocean. I cannot repeat here all the studies already cited in Wassmann et al. 2015 that are dealing with zooplankton and models due to the limitations of the letter format of the article, but the readers could refer to those references within. It is written that they do not use observations and they do not

investigate directly phytoplankton advection with observation. So, it suggests that phytoplankton studies use models and/or qualitative analysis.

page 2 line 83 : I wonder why the following publication is not cited: Optical Modeling of Spectral Backscattering and Remote Sensing Reflectance From *Emiliana huxleyi* Blooms
Griet Neukermans^{1*} and Georges Fournier²

The article above is about improving optical algorithm for the detection of EHux and is not related with our study. Our study does not focus on the improvement of the EHux tele-detection but use a standard algorithm of Particulate Inorganic Carbon (PIC) provided by the NASA Ocean Color (see Material and Methods).

page 3 line 110: Is this a citation?

Actually not, it was meant to represent the ADT field reconstructed from EOF 1 and 2. We recognize that this is misleading so we changed to ADT with EOF_{1,2} as indices (i.e. ADT_{EOF}).

page 3 line 121: Is the increase in advection a question of water volume or surface current velocity? If it is volume, who has determined the volume increase? A closer look at the work of Polyakov (including non-cited papers) may be fruitful

Question has been already answered above. But we would like to add some details in our answer.

We disagree with the reviewer that the authors found an increase volume transport. What they did find is an increase heat content which is different. The authors used ITPs boys and moorings to describe the hydrographic profiles (temperature and salinity) in order to investigate the vertical structure of the water column and to derive heat contents that affect the sea ice. From our knowledge, they did not use or describe any current measurement (despite the fact the ITPs are equipped with ADCP). They did not assess directly the advection of Atlantic waters. This is why we cannot use this work to contextualize the increase in surface ocean currents. However, the reviewer is right that other studies (such as those raised by reviewer #3) have actually estimated the volume transport at BSO with current meters and we admit our work have to better compare with previous studies. See our answers to his comments above for more details on the modifications provided to improve the manuscript in this way.

page 4 line 156: How is this barrier defined? Never heard the term.

Please refer to our answer to this question above.

Note that Carmack and McLaughlin (2011) also used the term thermal barrier.

page 5 line 178: The currents themselves. What does that mean? Does it mean that the volume of Atlantic Water has increased? And is this proven? As it is not temperature, then it must be volume flux. Or could it be the current in the surface is stronger than before?

The wording has been changed to “stronger surface currents”.

page 5 line 187: That is not generally true. Ehux blooms contain hardly any biomass, as compared to algae that are not easily detected from space. They do not play a major role in biogeochemical cycles, may be in geochemical ones. The fate of organisms in the Arctic Ocean does only marginally depend on Ehux.

As previously answered, we corrected to 'geochemical cycles'. See our answer above. Their role is crucial not necessarily because they contain biomass but because they are calcifying organisms that affect the carbon cycle.

page 5 line 188: Atlanticlike environment? Define.

Yes. We actually meant 'Atlantic Waters'. We have replaced accordingly as explained in our answer above.

page 5 line 192: Take note of Wassmann et al. 2015

Yes, Wassmann et al (2015) has been added.

page 5 line 193: Thriving with currents? Never heard this before. Citations?

Yes, we meant 'drift'. This concern was also answer before and corrected throughout the manuscript.

page 5 line 195: Any bloom is determined by growth conditions, physical condition AND grazing. One time a bloom may take place because the lack of grazing, the second time it may take place because growth is greater than grazing. Your ideas of phytoplankton blooms are simplistic.

Absolutely. Please see our answer above.

page 5 line 202: What is a short solar cycle? Define. The sun shines almost all the time from March to September. Is that short?

Please see our answer above.

page 5 line 207: The interior shelves start somewhere east of the Kara Sea. They are far from the spread of Atlantic water as yet. The interior shelf is wide and shallow and the Atlantic water will stay close to the shelf break. Or?

Yes, this is true that the interior shelves are still not affected by AW and that the Atlantification will first concern the shelf slope. If the "Atlantification" continues, the Barents Sea can become completely Atlantified (Lind et al. 2018), and that AW will spread at surface or sub-surface along the interior shelf slopes. This could be done through different processes (Polyakov et al. 2017) : (1) surface intrusions through the AW outflowing gateways (passage south of Novaya Zemlya or between Novaya Zemlya and Franz Josef Islands, BSX sensu Smedsrud et al. 2013). (2) increase penetration and shoaling (along with

weakening pycnocline) of AW along the shelf break. Some branches could potentially in those conditions recirculate into the shelf and re-surface.

We copy paste again the new version of the paragraph where we reworded following Carmack and Wassmann (2006), and we specified that the increased advection occurs along the slope:

“These conditions⁵⁷ would likely be met in the **Eurasian interior shelves** of the Arctic Ocean³⁰ (i.e. Kara, Laptev, and Siberian seas) where surface waters are warming and freshening⁵⁸. If the increase in advection along the **shelf slope** continues in the future, we can expect Ehux to become a permanent resident of the newly “Atlantified” Eurasian interior shelves, a similar scenario to that of the last interglacial⁴⁹”

page 5 line 212: The atlantification is northwards in the Barents Sea and eastern Fram strait. It is eastward north of Svalbard along the Eurasian shelf.

Yes exactly, what we meant here is that Ehux drift with the circulation toward the Arctic. The flow heading is indeed not always northward, and once on the shelf break North of Svalbard, it flows eastward following a cyclonic circulation. That's why we decided to use the term 'poleward'. Additionally, it remains true in our study region (i.e. Barents Sea especially and eastern Fram Strait).

page 5 line 215: It is difficult to accept the jump between Ehux and higher trophic levels when the manuscript totally ignores grazing.

We agree with the reviewer. We have included grazing as a constraint for EHux accumulation in surface waters as described in the previous comment. We hope that It may help accepting this part of the paragraph.

page 5 line 217: The decline in silicate and nitrate in the Atlantic water will have far more ecological consequences for the Arctic Ocean than Ehux. By far the most of the harvestable production is caused by silicate and nitrate.

The reviewer is absolutely right, the decrease in all inorganic nutrients (but in silicates particularly) will have important implications notably on diatoms. We have listed this important change referring to Rey (2012) in the last paragraph.

“By driving such a poleward expansion, advective processes could affect the entire marine ecosystems by shifting species⁶¹ and modifying interactions at higher trophic levels⁶². The concomitant decrease in silicate concentration in NAW⁶³ may also contribute to the success of non-silicifying and small phytoplankton such as *Ehux*⁶⁴.”

page 7 line 285: Why are these references here?

This was a mistake, sorry about that, this reference should have been deleted. It is now deleted.

Reviewer #2 (Remarks to the Author):

Dear reviewer #2,

Thank you for reviewing this article. We appreciated your comments which will for sure improve the quality of the manuscript.

In their manuscript, Oziel and coauthors test a very interesting hypothesis that the expansion of *E. huxleyi* in the Barents Sea is resulting from a recent increase of currents flowing from the Atlantic. The expansion of this species has been attributed to temperature or CO₂ concentration increase, but the transport hypothesis has never been discussed yet. It is natural to think that if the species does not stay in the Arctic during the winter because of darkness, they would have to migrate every spring and summer to those latitudes, and then because *E. huxleyi* is part of the plankton (meaning that it cannot move against a current) it has to be transported. It is indeed difficult to imagine *E. huxleyi* surviving the winter night at the latitude of the Barents Sea, and the self migration on hundreds of kilometers for an organism of 6 μm in diameter is impossible. The work is essentially based on satellite observations compared with Lagrangian simulations of currents. They show that the Atlantic current speed increase recently and therefore penetrates more the Barents Sea. The coccolithophore bloom migration by flow transportation is the easiest interpretation. The experiments indicate that *E. huxleyi* can be transported to those latitudes by the currents and therefore could explain the polar expansion of *E. huxleyi*. The Lagrangian simulations of the current in the different year, as presented in this work, surprisingly track the decadal expansion of the bloom. The demonstration is quite elegant and the text is well written. This is remarkable and therefore I recommend its publication in Nature Climate Change.

I have some comments and questions :

1- the term 'inoculation point' is not well adapted to this context, because it may be understood as a migration starts from just one point, which is not the case. I would like to read a clear description of the surface of origin of the bloom. Unfortunately, in Figure 9 in supplementary material, the blue dots do not appear on the map. 2- This figure should be redrawn (showing blue dots).

Thank you for raising this issue. The reviewer is right, the inoculum describes an area. We have now made sure that the 'inoculum' term is always followed by 'region', 'area' or 'patch'.

The reviewer is also right that we forgot to plot the virtual particles at the beginning of the experiments as blue dots. This is now corrected in former Fig. S9 (now Fig. S14).

2-A question arrive from Figure 10 in the suppl. Mat. : Why the simulated surface of the bloom (in red) is always smaller that the observed one (in grey). Could it be that the 4°C winter isotherm does not represent the best estimate of the winter edge expansion and instead it should be something colder like 3°C which should represent a larger surface of origin ? In any case, this figure is fascinating and should be placed in the main article.in replacement of Figure 4, or even 3 which is difficult to read.

Thank you so much. There is a misunderstanding. In Figure S10 (now figure S15), the fact that the grey (now reddish) shading (i.e. PIC concentration) is usually larger than the area described by virtual particles would be rather explained by: (1) the grey shading area contains a +50 km perimeter around each PIC pixel. (2) The threshold used here is quite low (0.002 mol/L) because to match up the locations of PIC and virtual particles, we are interested in the presence, not the abundance of coccolithophores. We also necessarily consider a perimeter around the pixels with PIC>0.002 mol/L, which is unavoidable for doing such a matchup. In order to improve the readability of the former Figure S10 (now S15), we now only show virtual particles originating from distance < 180 km from the coast.

We agree as well that the Figure 3 is difficult to read, especially Panels A and B. To improve it, we have changed the way we depicted the arrows. We have replaced the curved arrow by straight black arrows in addition with the amplitude in the background (see new Figure 3). Also, because the reviewer #1 has raised the issue that the recomposed ADT_{EOF} is difficult to understand, we replaced ADT_{EOF} by the full ADT signal in Fig. 3. We can still compare it with the re-constructed signal with the EOF but in Fig. S8 in supplementary material.

Concerning Figure 4, we spent a lot of time thinking about how to increase its quality because we totally agree it deserved improvements. We have hesitated a lot and we tried out many solutions. First, we tried to replace the figure 4 by the former figure S10 as the reviewer #2 advised. But in this case, we faced major issues. The information about the inoculum and the thermal limit were lacking for the interpretation of the data. We added the information but the figure became unreadable. Furthermore, it was a figure with 20 panels which, we believe, should be avoided in a letter format article. Also, we prefer to show the actual PIC distribution and not the contour with a +50km perimeter around each pixel such as in former Figure S10. We finally decided to keep up with Figure 4 comparing two contrasted years (1998 vs. 2015) but we tried to improve it the best we can to answer your concerns. Here the list of modifications undertaken:

- Increase size of particles and thickness of lines
- Lower the importance of the inoculum area by changing the blue brown.
- Increasing the contrast of the PIC and increase differentiation from the land by changing to a blue palette
- Draw NAC as arrows to illustrate the increase currents.

OLD VERSION:

NEW VERSION

We hope that those changes will answer your concerns.

3-The polar expansion of Ehux bloom has been first described by Winter et al., in 2014. It is surprising that this work is not mentioned in the main text. The only mention of it is in the supplementary material but with a wrong author ranking (Brown is placed as a first author) and with a wrong date (2013 instead of 2014). A description of this study should be discussed earlier in the paper, because it is well in line with this work and again is the seminal paper on that subject.

We are sorry about that. As a general comment, we had an issue with our mendeley bibliography reference manager which weirdly changed the first author. We have corrected the bibliography throughout the main text and the supplementary material. Winter et al. (2014) is now cited in the introductory paragraph as soon as we mention poleward intrusions of Ehux:

“is the major mechanism responsible for the recent poleward intrusions of southern species like EHux⁸”

8 – Winter et al. 2014

4- In general the reference list is poorly written. Many errors are found (see details below)

Yes, absolutely, we apologize for this. As mentioned by all reviewers, this comes from a bug in our reference manager. We have thoroughly corrected all references.

5-The term ‘niche’ is not appropriated because here it relates only to temperature; a niche being something more complex involving light, nutrients among many other parameters. Terms like ‘temperature limit’, ‘lowest temperature requirement for bloom development’, may be more adapted. The concept of niche in ecology is much broader than what is meant here (Also in the Supplementary material section, the description of the niche in term of temperature is too limitative and provide a very narrow temperature range - In the ocean, *E. huxleyi* has been found in temperature ranging from -1°C to 30°C. It blooms in the Atlantic at around 14-16°C. In culture its best development is at around 20°C.

We totally agree with the reviewer that an ecological niche refers to a multitude of environmental factors that are not restricted to temperature only but to many others (light, turbulence, nutrients, etc). In order to avoid confusion, we got rid of the mentions to the ecological niches by replacing it either with ‘thermal limit’ or ‘lowest temperature required for their growth’ (their = EHux) or just ‘inoculum area’.

For example:

“The 4°C surface isotherm (Fig. 4) is considered as a **thermal barrier** for EHux because it represents the **lowest temperature required for their growth**. The EHux blooms do not seem to follow **this thermal barrier** in summer and could expand further north than currently observed. However, a north-eastward expansion of the winter 4°C isotherm, which also delimits **the inoculum area** in our Lagrangian method, could play a critical role by reducing the distance between the EHux winter location and the Arctic domain.”

See tracked change manuscript for the other replacements throughout the manuscript.

We also agree that EHux can grow in a much broader temperature range especially in lower latitudes. The PhD thesis From Le Vu (2006), who specifically studied EHux ecological niche, gathered the existing literature about growth rate vs. temperature:

Figure 1.3 : Growth with temperature

Relation between specific growth rate and temperature for different strain of *Emiliania huxleyi* from different oceanic area. Growth rate from experiment to a variable Temperature (continuous line) and maximum growth rate corresponding to specific temperatures (constant). The bold curve corresponding to equation 1 proposed for SWAMCO model but to be calculated (see explanation how to calculate in the text III.B.2).

The Figure above from Le Vu shows, as the reviewer #2 mentioned, shows that the maximum growth rate occurs around 20degC. The 9°C upper limit is from a more recent results from Signorini and McLain (2009) who showed that there is a sharp decrease in PIC above 9°C in the Barents Sea:

However, those results are not based on growth rate experiments and only the minimum temperature allowing EHux growth does matter for this study. According to your remark, we have abandoned the non-necessary references to a maximum temperature for Ehux growth in the Barents Sea (see changes in Supplementary material 2.5).

6- What would be the conditions that would make Ehux bloom in the arctic. Would it be light with only a temperature limit ($>4^{\circ}\text{C}$)? That is a bit short view. Also if a cell stays in the same body of water and divide to reach bloom, why this body of water does not become sooner nutrient depleted ? Here the blooming cells would stay from March to August in the same water. In the rest of the ocean, blooms occur for few weeks and collapse. How does the multiplication by cell division is taken in consideration in this work ? Is there any type of mixing of the Atlantic water mass with local waters, that could help bringing new nutrients to fuel the bloom development. I do not see any mention of that.

We agree with the reviewer concerning the fact that 6 months is a long period of advection, and that therefore there is a necessity to explain how the blooming cells can survive for such a long period. In general, a water mass advected by the surface currents in the open ocean is typically subjected to submeso- and meso-scale dynamics ($\sim 1\text{-}100\text{kms}$, few days to weeks). Along the horizontal dimension, these processes tend to compress and elongate the water mass, creating convoluted filaments which can intrude far from their origin, and which can mix with the local water. Along the vertical dimension, submesoscale activity can induce important local vertical velocities, favouring the upwelling of nutrients at the surface, especially along slopes where the AW currents are. Therefore, these processes favour the local increase of nutrients, and can support the permanence of the EHux cells in these waters, even though they are adapted to oligotrophic conditions. The submeso- and meso-scale effect is clearly visible in Fig. S13, where the initial patch is deformed in several elongated filaments after the 6-months advection.

To better illustrate submeso and meso- scale features, we use the Lyapunov Exponents, often computed over finite size (FSLE, Boffetta et al. 2001, d'Ovidio et al. 2004, Shadden et al. 2005), which are classical diagnostic used to identify Lagrangian Coherent Structures.

As Lehanh et al. (2017) explained:

“Lyapunov exponents quantify exponential rates of deformation of water parcels. When computed backward in time, they permit one to identify the water masses that have undergone strong filamentation. This typically occurs along fronts and at eddy peripheries. A map of Lyapunov exponents is typically organized in ridges, which can be interpreted as regions where stirring is maximal. These identify tracer fronts and transport barriers. Other mathematical methods for computing Lagrangian coherent structures have also been proposed (Mancho et al. 2004, Mendoza & Mancho 2010, Haller & Beron-Vera 2012, Mundel et al. 2014).”

It is indeed known that the filamentation process, typical of turbulent dynamics at the (sub-) mesoscale, can enhance the contact and mixing of waters with contrasting properties

(Lapeyre and Klein 2006), favoring vertical movements (McGillicuddy Jr et al. 1998; Mahadevan et al. 2012; Levy et al. 2012).

The turbulence level in the Norwegian Sea is much higher than in the Barents Sea where the horizontal activity is less intense (see Figure below). It is interesting to note the low levels of stretching, stirring, fronts and filamentation due to the weak turbulence (low FSLE values) of the Atlantic side of the southern Barents Sea (see Figure below). Those patterns evidence low mixing activity than the rest of the EAC. On the one hand, the intense mixing in the Norwegian Sea can be a source of diversity through an inoculum of different phytoplanktonic species (d'Ovidio et al. 2010, Della Penna et al. 2018). On the other hand, it could entrain phytoplankton to a greater depth without sufficient available light for sustaining growth. Moreover, nutrient affinity of small cells like EHux are negatively affected by high turbulence levels (Lindermann et al. 2016).

Figure: Maps of (top) FSLE (day⁻¹) and (bottom) Particulate Inorganic Carbon (PIC, µmol L⁻¹) in August 2015. Strong filamentations are evidenced by large absolute values (dark areas). 9°C and 4°C isotherms are respectively shown in red and blue lines.

To put our answer in a nutshell: (1) water parcels are not necessarily isolated from the surrounding water masses and they experience alterations/modifications as they move along filamentation processes. Also, Ehux are specifically adapted to low nutrient concentrations. They would be outcompeted by diatoms or phaeocystis in a nutrient-rich waters. They usually bloom during oligotrophic conditions (low nutrient levels) while the minimum in nitrate

concentration is reached (~1 μM of nitrate in surface AW; Oziel et al. 2017) but still sufficient for Ehux development. (2) Also, their blooming is dependent on a multitude of environmental factors (not only light and temperature) that we tried to list in the last paragraph of the last section “A newly “Atlantified” and retreating marine Arctic ecosystem”:

“Ehux seems to be adapted to the low light, low nutrient, oligotrophic and highly stratified conditions of the NAW in summer⁵⁵ such that its expansion, growth and blooming in the Arctic Ocean will be limited at some point by those constraints (bottom-up) but also by the grazing pressure (top-down).”

In the previous section (Increasing NAW surface currents control the spatial distribution and poleward expansion of Ehux), we changed the formulation so that it is clearer that Ehux expansion toward the pole could be attributed to another factor than temperature:

“The 4°C surface isotherm (Fig. 4) is considered as a thermal barrier for EHux because it represents the lowest temperature required for their growth (Supplementary Result 2.5). However, the Ehux blooms do not seem to follow this thermal barrier in summer and appear to be constrained by another factor.”

We would like to add, just because this is interesting, that some recent studies showed that the plastic debris colonization can enhance the permanence of microorganisms at the surface, rafting them for several kilometers, and favouring the intrusion of non-native species. These has been shown in particular for high latitudes, which are the ones considered in the present study (Barnes et al., 2002). Coccolithophorides presence on plastic litter has already been observed (Carson 2013 et al.), and also E.Huxley one (Reisser 2015 et al.).

Additional References:

Boffetta G, Lacorata G, Redaelli G, Vulpiani A. 2001. Detecting barriers to transport: a review of different techniques. Phys. D 159:58–70

d'Ovidio F, Fernandez V, Hernandez-Garcia E, Lopez C. 2004. Mixing structures in the Mediterranean sea from finite-size Lyapunov exponents. Geophys. Res. Lett. 31:L17203

Shadden SC, Lekien F, Marsden JE. 2005. Definition and properties of Lagrangian coherent structures from finite-time Lyapunov exponents in two-dimensional aperiodic flows. Phys. D 212:271–304

Mancho AM, Small D, Wiggins S. 2004. Computation of hyperbolic trajectories and their stable and unstable manifolds for oceanographic flows represented as data sets. Nonlinear Process. Geophys. 11:17–33

- Mendoza C, Mancho AM. 2010. Hidden geometry of ocean flows. *Phys. Rev. Lett.* 105:038501
- Haller G, Beron-Vera FJ. 2012. Geodesic theory of transport barriers in two-dimensional flows. *Phys. D* 241:1680–702
- Mundel R, Fredj E, Gildor H, Rom-Kedar V. 2014. New Lagrangian diagnostics for characterizing fluid flow mixing. *Phys. Fluids* 26:126602
- Lapeyre, G., P. Klein, and B. L. Hua (2006), Oceanic restratification forced by surface frontogenesis, *J. Phys. Oceanogr.*, 36, doi:10.1175/JPO2923.11577
- McGillicuddy DJ Jr. 2016. Mechanisms of physical-biological-biogeochemical interaction at the oceanic mesoscale. *Annu. Rev. Mar. Sci.* 8:125–59
- Mahadevan A, Campbell J. 2002. Biogeochemical patchiness at the sea surface. *Geophys. Res. Lett.* 29:1926
- Levy M, Ferrari R, Franks PJS, Martin AP, Riviere P. 2012. Bringing physics to life at the submesoscale. *Geophys. Res. Lett.* 39:L14602
- d'Ovidio F, De Monte S, Alvain S, Dandonneau Y, Levy M. 2010. Fluid dynamical niches of phytoplankton types. *PNAS* 107:18366–70
- Lindemann C, Fiksen Ø, Andersen KH and Aksnes DL (2016) Scaling Laws in Phytoplankton Nutrient Uptake Affinity. *Front. Mar. Sci.* 3:26. doi: 10.3389/fmars.2016.00026
- Della Penna, A., Trull, T. W., Wotherspoon, S., De Monte, S., Johnson, C. R., & d'Ovidio, F. (2018). Mesoscale variability of conditions favoring an iron-induced diatom bloom downstream of the Kerguelen Plateau. *Journal of Geophysical Research: Oceans*, 123, 3355- 3367. <https://doi.org/10.1029/2018JC013884>
- Oziel, L., Neukermans, G., Ardyna, M., Lancelot, C., Tison, J-L., Wassmann, P., Sirven, J., Ruiz-Pino, D., and Gascard, J-C. (2017), Role for Atlantic inflows and sea ice loss on shifting phytoplankton blooms in the Barents Sea, *J. Geophys. Res. Oceans*, 122, 5121–5139, doi:[10.1002/2016JC012582](https://doi.org/10.1002/2016JC012582).
- Barnes D.K.A. (2002), Invasions by marine life on plastic debris, *Nature* volume 416, pages808–809.
- Carson H, Nerheim M, Carroll K, Eriksen M (2013) The plastic-associated microorganisms of the North Pacific Gyre. *Mar Pollut Bull* 75:126–132.
- Reisser J, Shaw J, Hallegraeff G, Proietti M, Barnes DKA, Thums M, et al. (2014) Millimeter-Sized Marine Plastics: A New Pelagic Habitat for Microorganisms and Invertebrates. *PLoS ONE* 9(6): e100289. <https://doi.org/10.1371/journal.pone.0100289>

7 The end of the discussion is quite speculative and also may appear in contradiction with the main findings : It is said that *E. huxleyi* could penetrate permanently to East in the southern shelves of the Arctic Ocean : the problem that is foreseen is that 1) the winter darkness, which prevents *E.huxleyi* to survive the winter (this is the first postulate of the manuscript), will not change in the future, 2) if this future yearly colonization by *E.huxleyi* of the Arctic Ocean is due to ocean warming, then one may wonder why it did not do that in the past since *E. huxleyi* is a eurythermal species.

The reviewer is right. Thank you for raising those issues.

- (1) We cannot say that *Ehux* will become a permanent resident, because of winter darkness and cold, as you say. We modified to “summer resident”.
- (2) Absolutely! There is evidences that *Ehux* did actually occupy, along with Atlantic Waters, the Arctic Ocean during the last interglacial.

“The geological record suggests that *E. huxleyi* was also present in the Arctic during diminished sea-ice conditions of past interglacial times (Backman *et al.*, 2009).” Winter *et al.* 2014

Baumann *et al.* 1990: “During the last interglacial, North Atlantic Current water and pack-ice distribution, comparable to ice conditions during stage 3 in the Fram Strait, enabled coccolithophorids to live up to at least 86°08'N.”

ACTION:

In the penultimate paragraph, we ended with:

*“If the increase in advection along the shelf slope continues in the future, we can expect *Ehux* to become a summer resident of the newly “Atlantified” Eurasian interior shelves, as previously revealed during the last interglacial^{49,60} ”*

Minor remarks :

Line 49 : NAW ‘carry most of the heat’ : imprecise and not clear

We recognize the imprecision of the sentence. The main source of oceanic heat in the Arctic Ocean comes from the warm North Atlantic Waters. NAW is therefore a ‘vehicle’ that transports most of the ocean heat from the lower latitudes toward the pole. We re-worded:

“... the North Atlantic Waters transport most of the ocean heat⁵ ...”

Line 81 : deepest should be replaced by something like ‘farthest’

Modification done

Line 114: +2mm s-1 yr-1 does not seem very much (like 172 m /day or 15 km in 3 months) : can you comment on this ?

Of course! Your calculation is absolutely right. We admit $2 \text{ mm s}^{-1} \text{ yr}^{-1}$ may appear low but it means that during a 6-months drift (such as in our Lagrangian experiments), particles covers about +30 km each year (equivalent to your +15 km in 3 months). Imagining a constant increase of $2 \text{ mm s}^{-1} \text{ yr}^{-1}$ (maximum) over the entire time-series (24 years), it would represent an maximum increase in distance travelled by about 720 km which is far from negligible. In comparison, the study area is 2200 km long (extends from 60degN to 80degN; $\sim 2200 \text{ km} \sim 3 \times 720 \text{ km}$).

Line 156 : why Ehux blooms do not follow the surface thermal barrier when later it is said that the thermal barrier results from Atlantic water advection.

At surface, in summer, the water is also warmed by solar radiation (not only due to AW advection) which extends the 4degC isotherm farther north than AW. It is usually fresher water (see for example Oziel et al. 2017).

Line 159 : 'could play a critical role by' could be replaced by 'help'

Thank you, this correction was included.

Line 163 & 164 : 'at the same location' seems in contradiction with determining the inoculation region. A clarification is needed.

Sorry, we meant that for EXP2, we used the same inoculation area for each year. We replaced 'location' by 'inoculation area'.

Line 177: Winter et al., 2014 should be quoted... and also (as I mentioned earlier) it should be discussed in the introduction since it is the first mention of Ehux Polar Expansion.

Of course, this is done as described in our previous comment as well as in the introductory paragraph (abstract).

Line 202 : the reference is not written well in the reference list (see below).

Done

Line 204 : 'short solar cycle' - No - short night or long days but the solar cycle is not shortened : 24 hours in most of the cases or 1 year at the north pole .

Absolutely. Together with the issues pointed out by reviewer 1, we re-worded as it follows:

"the survival of Ehux would require adaptation to rapidly decreasing solar radiations in late summer⁵⁷"

Line 205 : 'warm' -> relatively warm : Arctic is not tropical :-); 'Rest' - not appropriated.

Sure! Do you agree we use the term 'temperate' instead of relatively warm ?

To avoid confusion, we simply decided to delete "rest". Because of other concerns raised by reviewer #1, the sentence becomes:

"Advection with the surface currents, Ehux will have to survive during the travel (e.g. avoid grazing^{52,53}, subduction under the polar mixed layer in Fram Strait⁵⁴...) until finding more favourable blooming conditions in the BS in summer¹⁸"

The reference list is poorly written :

For example :

Ref 5. Beszcynska-mo for Beszcynska-Moëller and Marine Science written twice, and no pages.

Ref 13 <<i&.... In the title

Ref. 28 BIRKENES & E ??? Who is E ?

Ref 44 the beginning of the Abstract is written instead of the title and journal.

Ref 26 of Supplements : Author are not in order and a wrong year (should be 2014).

We are sorry about those lacks. We have thoroughly corrected the references. We encountered major 'bugs' with our bibliography software. We made sure that everything is corrected manually now.

Reviewer #3 (Remarks to the Author):

Review, Oziel et al 2019: Faster Atlantic currents drive poleward expansion of temperate marine species in the Arctic Ocean.

This paper bases its conclusions on two fundamentals:

A velocity field estimated from satellite altimetry for the time period 1993 to 2016

A distribution of *Emiliana Huxleyi* estimated from satellite observations of Particulate Inorganic Concentration for the same time period.

There is a change in the leading edge of the PIC in the Barents Sea during that period: northeastward. They use the (changing) velocity field to trace lagrangian particles and conclude that the moving leading edge can be explained primarily by an increase in velocity

of the Atlantic Water and not so much by the increase in temperature of the Atlantic Water, as was previously assumed.

The main problem I have with this manuscript is that the authors do not investigate the velocity field as thoroughly as they ought to. Most investigations of the heat budgets of the Barents Sea and the Arctic Ocean rests on current meter measurements at the Barents Sea opening, of which there are now nearly 20 years of data. And even before 1997, there were several published estimates of that transport. I have tried to collect them in the list below.

Timofeyev, 1963: 2 Sv Atlantic Water eastward, 0.9 Sv westward return flow, net east: 1.1 Sv
Blindheim, 1989: 3.1 Sv Atlantic Water eastward, 1.2 Sv return westward flow, net east: 1.9 Sv.

Mauritzen, 1996: net east: 1.6 Sv

Skagseth, 2008: 2.0 Sv Atlantic Water eastward (time period 2003-2007).

Skagseth et al. 2008 (Volume and heat transports to the Arctic Ocean via the Norwegian and Barents Seas.): 1.8 Sv net east, time period 2000-2006, with a positive trend from 1 Sv in 2006 to 2 Sv in 2006.

Smedsrud et al, 2010: 2 Sv Atlantic water eastward, 1.2 Sv Norwegian Coastal Current Water eastward, 1.2 Sv westward return flow, net east: 2 Sv. (time period 1997-2007)

Årthun et al 2012: 2.3 Sv eastward, time period 2000-2009, with a positive trend.

As one can tell from these numbers, it is possible to argue that the volume flux has increased during the 2000s [but the changes are not much larger than the expected error bars of the measurements]. Nevertheless, the authors have not made these comparisons. The two estimates showing positive trends (Årthun et al and Skagseth et al) display completely different time evolution, and neither one resembles this manuscript's figure 2B. Without a thorough discussion of their velocity findings, in the context of previous publications and the observations made in the area, this reviewer has problems accepting the velocity results upon which the paper is based.

Dear reviewer #3,

Thank you for reviewing this article.

We want to thank the reviewer for raising these issues. These comments made us realize that we had not properly described the performed validation towards observed currents, as well as that we should more carefully distinguish between surface/upper layer currents and total volume transports. In addition, we have evaluated the uncertainties in the derived surface ocean currents, and how this impact on the Lagrangian method. Clarifications and better explanations had been added to the main text and supplementary material, and several new analyses are presented in the supplementary material. More details on each of the issues raised by the reviewer is given below.

a. Validation of altimeter data in the literature

Surface geostrophic velocities fields from altimeter data have been improved and with a special focus on the Arctic regions in recent years. The dataset has been validated in the Norwegian Sea and the Barents Sea and has shown good agreement with observations.

This is described in Material and Method in the main document, and with more details in the Supplementary Material and Method 1.1 with relevant references.

We underline that the use of altimetric currents to infer temporal trends in current intensity has already been employed in literature, and does not constitute a novelty of our work. In particular, we highlight the works of Backeberg et al., 2011, Elipot & Beal, 2018, which revealed an increased Agulhas current thanks to the use of altimetric velocities.

b. Validation in this study

We have validated the satellite-derived geostrophic velocity with *in situ* current meters in Supplementary Results 2.1 (Figure. S3). The current meter data is from 50 m depth in the BSO, and is the same employed in the literature you collected (updated from Ingvaldsen et al. 2004a). We show that the retrieved velocities were in relatively good agreement with uppermost current meters in the BSO. Note the current meter **data series has been updated up to 2016** during the revision (vs. 2012 in the previous version). We realize that the validation was not properly mentioned in the main text during the previous version, and we have now added the following sentence to the Material and Methods section: “For this study, the derived surface geostrophic velocities were validated towards long-term *in situ* measurements in the Barents Sea (Supplemental Result 2.1).”

c. Estimation of volume transports

Although geostrophic velocities caused by sea level changes associated with Ekman transports are important drivers for the Atlantic Water velocity field in the there is also a baroclinic component of the flow associated with the internal density gradients (Ingvaldsen et al. 2004a). Reliable estimation of the Atlantic Water flow in the full water column can be obtained from altimeter data, but then in combination with hydrography. Such an approach is beyond the scope of this work focusing on transport of *Emiliana huxleyi* occurring in the upper water layers. Thus, the presented altimetry derived geostrophic velocity fields are to be taken as representative of the surface mixed layer flow of Atlantic Water and not for the full water column. Consequently, no volume transport estimates are calculated to be put in the context of previous publications of transports. This is now explained in the Supplementary Material, and we carefully rephrased the manuscript to emphasize that we focus on the surface/upper layer ocean currents only.

d. Trends in BSO

Concerning trends in the BSO, an additional analysis was performed (see Figure S4 in Supplementary Result 2.1) in which we showed the temporal evolution of the BSO eastward velocities. One main finding is that most of the increasing trend in the altimeter derived geostrophic currents at BSO occurred outside the mooring locations, north of M5 (Figure S4). This explains why the trend is not well captured by moored current meters at those locations and value the support of geostrophic derived trend in the region (see Supplementary result 2.1).

e. Uncertainties in the derived surface ocean currents and impact on the Lagrangian method

Ocean velocity fields inferred from altimetry have inherent uncertainties:

- (1) do not resolve sub-mesoscale and small mesoscale activity (1–50 km, Xu & Fu 2012)
- (2) have smoother mesoscale features because of tracks interpolations (Le Traon et al. 1998, Pascual et al. 2006)
- (3) do not estimate ageostrophic currents, Ekman transport and vertical processes (Capet et al. 2008; Sudre & Morrow 2008)
- (4) have errors associated with the vicinity of the coastline (Cipollini et al. 2014).

Additionally, there are several other sources of error (mean dynamic topography, correction of tides and inverse barometer, correction of the orbit, etc).

To assess the robustness of the Lagrangian technique to the presence of these errors, one of the best-case methodology is constituted by the use of a Monte-Carlo analysis. In this method, for each advective timestep, the position of the particle is affected by the local error on the velocity field. Unfortunately, maps of geostrophic velocity error do not exist yet and are still in development. Therefore, we decided to use a constant diffusion term (D in units of m^2/s), representative of the horizontal turbulent diffusion, added to the Lagrangian advection. Adding such a term means simply imposing an error on the velocity field which is uniform over the whole domain studied. This comes from the fact that D can be written as a function of the error on the velocity field, as we described in Supplementary Result 2.4.

To assess the value of D to be used, we utilized two complementary approaches:

- 1) On one side, we adopted the estimation methodology of Leeuwenburgh et al., obtaining in turn a diffusion coefficient which ranges between 26 and 30 m^2/s , according to the different years.
- 2) On the other side, we used a novel, experimental estimation of the velocity error provided by Aviso (pers. comm. Marie-Isabelle Pujol), which was available only for 2017. Using Eq (2) (Supplementary Result 2.4), the error was converted to the equivalent diffusion (see below Figure. 1).

Figure 1: Daily timeseries of Diffusion (m^2/s) based of geostrophic velocities errors in 2017 averaged over the studied area.

As it is possible to see in Fig.1, the diffusion does not change consistently in time, ranging between 7.5 and 11 m^2/s . We also computed the yearly averaged diffusion for each point of the domain, which is depicted in Figure 2 below.

Figure 2: Spatial distribution of the yearly averaged Diffusion (m^2/s) based of geostrophic velocities errors in 2017.

As it is possible to infer from the plot, only a relatively small region is affected by values of diffusion around $20\text{m}^2/\text{s}$ (0-15 in longitude, 68-70 in latitude, Lofoten area), while higher diffusion values are present only along Russian shores for example in the Kara Sea, which are not affecting our advection. The rest of the domain presents instead smaller values. Diffusion daily maps were analyzed to determine eventual time windows and regions of enhanced diffusion. However, these maps (not reported) presented a pretty uniform temporal trend, with patterns very similar to the one depicted in Fig. 2. This is in accordance also with the mean temporal trend reported in Fig. 1.

To obtain a more robust analysis, we decided to use the higher diffusion estimates, in that we adopted the values obtained through the Leeuwenburgh et al. method, which are on average 3 times higher than those obtained with the experimental product.

Once the diffusion term was determined, the Lagrangian experiment 1 (EXP.1, variable inoculum area, variable currents) was performed 1000 times. The analyses on the 'leading edge' were compared with those obtained from the experiment without diffusion (EXP.1). The results, presented in Supplementary Result 2.4 and associated Supplementary Figure S10, show that the Lagrangian method is very robust to the addition of noise representing a realistic error estimate. The expansion of the leading edge from EXP.1d is very similar to the non-diffusive scenario (trend = +380 kms vs +424 kms originally for EXP.1), with very low standard deviations.

But assuming that this manuscript's velocity fields are representing reality, the next "problem" is that the advection of EHUX is over-explained by the changes in velocity and temperature (see figure 5). Including both effects, the particles are advected 100 km further than they should have been (which is 326 km). Including only temperature effects, the particles are advected 140 km short of the target. I.e. the temperature explanation gets us 187 km towards the target, and one only needs another 140 km to reach the target. Therefore it is misleading to state (as is done in the abstract) that "we further demonstrate that bio-advection, rather than water temperature as previously assumed, is the major mechanism responsible for the recent intrusion....".

Thank you for raising these issues that clearly need to be discussed more clearly. Our results are based on the following:

- (1) As stated before, the parameters between EXP.2 and EXP.3 are not linearly independent and therefore, EXP.1 may not be equal to EXP.2+EXP.3. In order to reach this equilibrium (EXP.1=EXP.2+EXP.3), the distance from the coast has been artificially fixed to 180 km (see in supplementary results 2.4). In a sensitivity test (Fig. S11), we varied the distance from the coast from 100 km to 400 km every 1 km. At 180 km, the equilibrium EXP.1 = EXP.2+EXP.3 was found. But the criterion could have been to find the following equilibrium EXP.1 = OCEAN COLOR PIC, in which case the distance from the coast would have been around 100 km. However, in this case, the equilibrium with EXP.2 and EXP.3 is lost. This decision is an arbitrary choice we completely assume. It gives us the mean to quantify the respective roles of currents and temperatures, but on the other hand, we cannot really compare EXP.1 vs. Ocean color PIC.

This is why we decided to independently compare EXP.2 with EXP.1, and EXP.3 with EXP.1, because EXP.1 remains our reference to derive the relative contribution.

- (2) Temperature and currents are not linearly independent. Our intention in doing such an experiment is to derive the relative contributions of currents and temperature distinctly. However, those two parameters are not fully linearly independent because the increase in temperature is either due to (1) increase heat flux (either from temperature increase of the AW or increased AW advection) or to (2) increased warming due to solar radiation. In other words, the increased sea surface temperature itself may be partly explained by an increased AW advection (increase in AW currents) so the role of the currents here are very probably underestimated. As concluded by Raj and colleagues: “whereas the contribution of heat advection becomes as important as the ocean-atmosphere heat exchange at interannual time scales”. By contrast, the increase temperature has a very limited effect on geostrophic currents at this non-seasonal time scale (see Supplemental Results 2.3 - estimation of steric effects). To sum up, we are confident the 56% contribution from currents is rather underestimated than overestimated (the opposite that the reviewer proposed). Another reason is that surface geostrophic velocities tend to be underestimated (Volkov and Pujol, 2012).

We however realize we need to relax our claim in the abstract and to change the sentence to:

“We further demonstrate that bio-advection, rather than water temperature as previously assumed⁷, is a the major mechanism responsible for the recent poleward intrusions of southern species like *Ehux*⁸”

References:

Backeberg, B.C., P. Penven, M. Rouault. 2012. Impact of intensified Indian Ocean winds on mesoscale variability in the Agulhas system *Nature Climate Change*, 2 (2012), pp. 608-612

Berg, H. C. (1993). *Random walks in biology*. Princeton University Press.

Capet X, McWilliams JC, Molemaker MJ, Shchepetkin AF. 2008. Mesoscale to submesoscale transition in the California Current System. Part I: flow structure, eddy flux, and observational tests. *J. Phys. Oceanogr.* 38:29–43

Cotte, Cédric, d'Ovidio, Francesco, Chaigneau, Alexis, Lèvy, Marina, Taupier-Letage, Isabelle, Mate, Bruce, Guinet, Christophe, (2011), Scale-dependent interactions of Mediterranean whales with marine dynamics, *Limnology and Oceanography*, 56, doi: 10.4319/lo.2011.56.1.0219.

Elipot, S. and L.M. Beal, 2018: Observed Agulhas Current Sensitivity to Interannual and Long-Term Trend Atmospheric Forcings. *J. Climate*, 31, 3077–3098, <https://doi.org/10.1175/JCLI-D-17-0597.1>

Leeuwenburgh, O., and D. Stammer, Uncertainties in altimetry-based velocity estimates, *J. Geophys. Res.*, 107(C10), 3175, doi:10.1029/2001JC000937, 2002.

Legras, B., Pisso, I., Berthet, G., and Lefèvre, F.: Variability of the Lagrangian turbulent diffusion in the lower stratosphere, *Atmos. Chem. Phys.*, 5, 1605-1622, <https://doi.org/10.5194/acp-5-1605-2005>, 2005.

Le Traon, P., F. Nadal, and N. Ducet (1998), An improved mapping method of multisatellite altimeter data, *J. Atmos. Oceanic Technol.*, 15, 522–534.

Pascual A, Faugere Y, Larnicol G, LeTraon P-Y. 2006. Improved description of the ocean mesoscale variability by combining four satellite altimeters. *Geophys. Res. Lett.* 33:L02611

Sudre J, Morrow RA. 2008. Global surface currents: a high-resolution product for investigating ocean dynamics. *Ocean Dyn.* 58:101

Volkov, D. L. & Pujol, M. I. Quality assessment of a satellite altimetry data product in the Nordic, Barents, and Kara seas. *J. Geophys. Res. Ocean.* 117, 1–18 (2012).

Xu Y, Fu LL. 2012. The effects of altimeter instrument noise on the estimation of the wavenumber spectrum of sea surface height. *J. Phys. Oceanogr.* 42:2229–33

Let me be clear: I find the paper intriguing, and possibly publishable. But the authors need, in my opinion, to 1) make a better case for their velocity field, and 2) to reformulate their major findings to align with the actual results.

We thank you very much for your review. We have really appreciated your comments and we are convinced all those additional analyses have improved the quality of the manuscript.

Best regards,

Laurent Oziel and co-authors.

Reviewers' comments:

Reviewer #2 (Remarks to the Author):

I review for a second time the manuscript by Oziel and colleagues. The revisions produce in this second manuscript appear to be extremely well done. The manuscript is easy to read, and I do not have any problem with it. The authors have answered well to all my remark (except one, see below). I read with attention how they answered the question of Review 1: apparently the reviewer was not as positive I was on the 1st manuscript, most of his/her remarks are related to the consideration of older work. The author in the second version produced a noticeable effort in taking carefully his/her advice. I shared the same feeling about 'thermal barrier,' and on this point they do not answer completely, as they just explain in more detail what they mean by thermal barrier. I will come back to that later. For the rest they answer positively to all questions of Rev. 1.

There is only one minor point for which I am worrying on the meaning of the 'thermal barrier.' They write:

"The 4 °C surface isotherm (Fig. 4) is considered as a thermal barrier for *Ehux* because it represents the lowest temperature required for their growth (Supplementary Result 2.5). However, the *Ehux* blooms do not seem to follow this thermal barrier in summer and appear to be constrained by another factor. For example, the north-eastward expansion of the winter 4 °C isotherm, which delimits the inoculum area in our Lagrangian method, could contribute to reduce the distance between the *Ehux* winter location and the Arctic domain."

In the supplementary material is said: 'Despite the fact it is exceptionally apt at surviving in cold conditions ($T = 0\text{ °C}$), most studies (ref. 35–37) agree that its temperature niche is defined by temperatures above 3 °C'

I checked Ref.35 (Balch 2005) and Ref.34 (Paasche, 2001). Because I think there is a shift of the references 34 in the list is referred as 35 in the text.

Paasche mentions a limit at 1 °C and Balch does not speak of 3 °C. I did not find Ref.36 (Vu, B.Le) because the reference is not complete.

There is no agreement on that growth limit.

Again later, it is said 'the studies (ref.40,41) compiling the most extensive field collection of *Ehux* data in which 4 °C was the coldest temperature observed that was associated with non-zero growth rates of this phytoplankton.'

Ref 39 (=40 in the text) mention a limit at 4 °C, but from culture not in situ. Growth rate was given from literature ('*E. huxleyi* growth rate data were obtained from culture experiments detailed in the

primary literature'). It is very difficult to grow *E.hux* or any other organism at its limits of adaptation. The collection of strains, in Ref. 39 are in the range of 5 °C-25 °C. Therefore it is not surprising that the only attempt at 4 °C was not successful.

I did not find any reference on low temperature in Ref 40 (=41 in the text)

It is not possible that there is a lower growth limit at 3 or 4 °C: if below 3 °C *E.huxleyi* would not grow, how could we find this species at 0 °C (I collected it at that temperature in several occasions many studies show that) ? In conclusion, if *Ehux* is found at lower temperature than 3 °C it has to grow below 3 °C !

This is a minor point raised by Reviewer 1 and myself. It is minor because it does not change the manuscript. It is just a problem of semantic. 'Thermal barrier' and 'Ehux minimal temperature for growth' could be changed by something like 'Ehux minimal temperature for regular growth.'

Beside this small problem, I found the paper extremely interesting well written, and highly important for our understanding of the Arctic changing ecosystems.

Reviewer #4 (Remarks to the Author):

This paper claims that an increase in the intensity of currents in the North Atlantic can explain the shift towards the Arctic of the coccolithophore *Emiliana huxleyi* (*Ehux*). The authors use as input data a time series of 25 years of standard satellite altimetry to estimate surface currents, two satellite-derived products (sea surface temperature) and particulate inorganic carbon concentrations derived from satellite ocean color as a proxy for coccolithophore biomass. In addition, a lagrangian code is run to estimate particle trajectories.

The results are to my knowledge novel and could be of interest to others in the community as there is an attempt to show the potential impact of surface currents on a marine species. The paper is well written and follows a logic flow but unfortunately it is far from being convincing specially for a high-impact journal such as *Nature Communications*.

I present below my general comments:

1) Results are not well supported by the data because they do not demonstrate the real relation between currents and coccolithophore. What they are showing is just a correlation but not a demonstration that the primary driver of the *Ehux* dynamics are surface currents. I am not an

expert in this area in Eflux dynamics but I suspect the problem is highly more complex and the approach followed in this paper is oversimplified.

2) I have serious concerns on the appropriateness and validity of the data used and the statistical analysis performed. In particular, what is the robustness of the trends shown in figure 5? It is necessary to perform an error analysis as a trend without an associated error is meaningless.

3) Even if altimetry is a major tool for studying ocean circulation and has revolutionized our understanding of mesoscale variability, its reliability in an area so close to the North pole and to the coast as in this study is doubtful. Due to the specific orbit inclination of TOPEX/POSEIDON and the Jason series (the reference high accuracy altimeter missions), at latitudes higher than 66°N, the amount of data is very limited as it only comes from other missions such as ERS, Envisat, with lower accuracy and with a temporal repetitivity of 35 days, even if with a higher spatial resolution. Regarding the coastal band, it is well known that standard altimeter grided products in the coastal zone are adversely affected by land contamination with degraded range and geophysical corrections (see for example Vignudelli et al. 2011). A comparison with in situ data from currentmeters (for example) is required in order to show the potential validity of altimetry in this area.

REFERENCE

Vignudelli, S., Kostianoy, A. G., Cipollini, P., and Benveniste, J. (eds). (2011). Coastal Altimetry. Berlin: Springer-Verlag, 578. doi: 10.1007/978-3-642-12796-0

Reviewer #5 (Remarks to the Author):

The editor asked me to assess if authors addressed properly technical points related to usage of altimeter data that were raised by a reviewer who was unable to help at this round.

In this paper, the authors use satellite-derived altimetry observations to study the North Atlantic current surface velocities in the Arctic Ocean over the last 24 years. Their interest is at the long-term temporal scales (interannual and trends).

During the last decade, standard altimeter data underwent to several reprocessing campaigns with the objective of homogeneous different missions and inclusion of new/improved corrections, better inter-calibration, updated vertical references in order to estimate sea level anomalies. In parallel, some scientists made validation of these products in specific regions to assess their accuracy for

exploitation in oceanographic studies. The actual multi-mission products available via the Copernicus Marine Services (<http://marine.copernicus.eu/>) are benefiting of these enhancements and represent the state-of-the-art to study global and regional ocean circulations.

Therefore, this paper is one excellent example of fully exploitation of one of these products. It is also important to highlight that there are ongoing efforts in extending the satellite-based sea level record in challenging domains (coastal zone, inland waters and sea ice regions). Moreover, the European Space Agency Climate Change Initiative (CCI) project on “Sea Level” has reprocessed altimeter data to provide satellite-based sea level products to study climate-scale variations of sea level globally and in the coastal zone. The global product (available via ESA) provides monthly means of Mean Sea Level Anomaly @¼ degree during 1993-2015 using TOPEX/Poseidon, Jason-1, Jason-2, ERS-1, ERS-2, GeoSat Follow-On (GFO), Envisat, SARAL/AltiKa and CryoSat-2 with special care of editing, cross-calibration, homogeneous corrections, removal of global and regional biases, homogenization of long-spatial-scale errors, monthly optimal interpolation gridding. Major details at Legeais, J.-F., Ablain, M., Zawadzki, L., Zuo, H., Johannessen, J. A., Scharffenberg, M. G., Fenoglio-Marc, L., Fernandes, M. J., Andersen, O. B., Rudenko, S., Cipollini, P., Quartly, G. D., Passaro, M., Cazenave, A., and Benveniste, J.: An improved and homogeneous altimeter sea level record from the ESA Climate Change Initiative, *Earth Syst. Sci. Data*, 10, 281–301, <https://doi.org/10.5194/essd-10-281-2018>, 2018.

The authors in this manuscript use the satellite altimetry product generated by the Sea Level Thematic Assembly Centre (TAC) and distributed by the Copernicus Marine and Environment Monitoring Service (CMEMS). This is a gridded product with a spatial resolution @¼ degree with 1-day temporal resolution during 1993 to 2016. It is generated using Sentinel 3A, Jason 3, HY2A, Saral/AltiKa, Cryosat 2, OSTM/Jason 2, Jason-1, Topex/Poseidon, Envisat, GFO, ERS-1/2). The satellite altimetry system provides Sea Level Anomaly (SLA) that takes into account the variation of the Sea Surface height (orbit minus range corrected) around a vertical reference (Mean Sea Surface).

SLA from CCI and TAC products are not exactly the same as the processing chains are different. The authors use the SLA TAC because their interest is not in sea level but in ocean velocities. Therefore, they need daily data sets to use in synergy with the Mean Dynamic Topography (MDT) provided by the CNES-CLS13 dataset produced by the Collecte Localisation Satellites (CLS) Space Oceanography division and distributed by Aviso to compute the Absolute Dynamic Topography (ADT=SLA +MDT) from which to derive the velocities.

Having said that, the authors estimate an altimetry-based velocity field for the time period 1993 to 2016. The main point raised by the previous reviewer concerned the assessment of this field in terms of accuracy (the so-called validation using independent data sets) over the area of investigation. This is an important aspect of the analysis in order to support the acceptance of the scientific findings (i.e. that the observed increase of the water volume during the 2000s is error free and volume changes are not much larger than the expected error bars of the measurements).

The authors agree that the computed velocity fields need to be supported with more explanations and quantifications about their accuracy.

In the manuscript, section “Material and Methods”, the authors briefly describe the altimetry data sets and following reviewer’s comment a paragraph about validation is added with a pointing to supplementary for more details (section “1.1. Details on the altimetry dataset”).

I think the paragraphs need some rewording to avoid confusion in the reader. In the manuscript, the authors have just to specify the products they use, i.e. SLA TAC, MDT with source, spatial and temporal resolution. Then the authors explain the method ($ADT=SLA+DT$) and derivation of geostrophic velocities. For example the sentence “excluding pixels with error estimates (percentage of the signal variance) higher than 50” that is a technical detail has to be moved in the supplementary part, where I would remove “The Sea Level data are corrected for instrumental errors, inverted barometer, tides and tropospheric effects” because it is too generic (a table with specific sources would be necessary or a product reference) and incomplete (e.g., we need to correct also for ionosphere, SSB, DAC). DAC (Dynamic atmospheric correction) that includes wind effects is necessary to reduce the aliasing of the high-frequency sea level variability in altimetry data, especially in coastal regions. I also suggest to specify “The Sea Level Thematic Assembly Centre (TAC)” when CMEMS is mentioned and explain the exact product that is used (as there are different products, e.g., NRT, reprocessed, global, regional specific).

Then the authors have to convince the reader that the dataset they use is of quality and accurate. Concerning the quality, as highlighted by Volkov (2012) there is a problem of altimetry coverage. The advantage of the SLA TAC (compared to the data set used by Volkov) is the usage of CryoSat-2 that envelopes the Arctic region up to 220 km from the pole, while all the other missions have inclination allowing to cover the Arctic Ocean not closer than 950 km (81.5 deg North). Moreover, there is the problem of the seasonal presence of sea ice for which the altimetry data in the study region have gaps (see Volkov 2012, Figure 2). I think the authors do not provide any clarification about the new product with reference to satellite coverage (CryoSat-2 was launched in 2009 so what about before?) and presence of sea ice that is an indicator of the quality of data used.

A point that is not mentioned in the methodology is how geostrophic velocities are computed. As showed in Liu Y., Weisberg R. H., Vignudelli S., Roblou L., Merz C. R.: Comparison of the X-TRACK Altimetry Estimated Currents with Moored ADCP and HF radar Observations on the West Florida Shelf, in Special Issue “COSPAR Symposium”, Journal of Advances in Space Research, 50 (8), 1085-1098, doi:10.1016/j.asr.2011.09.012, 2012, white noise in altimetry data can affect the accuracy of the slope estimate from SLAs. Are the authors using a filter? I am stressing about providing details about processing because we need to ensure that other scientists can reproduce the work

Concerning the validation, the authors just point to references, in the manuscript. In my opinion, the authors have to provide some quantitative figures of the validation results from bibliography (Norwegian Sea, the whole EAC, Barents Sea) and from their independent validation analysis against

long-term in situ measurements in the southeastern Barents Sea. The readers will be then pointed to the Supplementary Material to better see in detail where the validations were made, which altimeter data were used, which in situ data were used. The reader also expects to have here a discussion of the statistics from the comparisons between altimetry and in situ data. I agree to mention Volkov (2012) who is not validating velocities but sea level anomaly, however, these figures will be supportive of the new product (that is certainly better than previous one). Ray et al (2018) is another important reference as they use the same product and the comparison is about altimeter-derived velocities against those measured by drifters.

The authors also added a validation against current meter measurements from 50 m depth in the Barents Sea Opening (section 2.1. Validation of satellite-derived geostrophic velocities toward in situ instruments at BSO in supplementary material). In situ data are averaged daily to remove tides and then averaged further weekly. A 4-weeks running mean is also applied to the time series. One plot shows the comparison at moorings with statistics and another one is showing velocity trends. Here there are some points that need to be clarified: 1) a map showing the mooring locations is necessary; 2) which altimetry grid point is selected for comparison (closest? radius?); 3) altimeter-derived velocities are daily (Is the running mean applied to this data set? If not, why?).

The authors will clarify differences between altimetry-derived and in situ data velocities in the answers to the reviewers. Ocean velocity fields inferred from altimetry exhibits various uncertainties. The altimeter measures both the barotropic and baroclinic signals. Altimetry provides an integrated measurement of the water column. Sea level is transformed in velocity using the geostrophic approximation. In order to avoid aliasing, high frequency wind effects have to be removed too. Current meters measure pointwise real velocities. Disagreements can be explained in terms small-scale variability, ageostrophic behaviour, increased variance going closer coastline. There are also several errors related to the altimetry system (error budget is at cm level in open ocean). By adding the Ekman velocity component to altimeter-derived velocity, the agreement is much better (see e.g. Liu et al. (2012) for a thorough comparison). This reference can be cited to reinforce the potential of altimetry to derive ocean velocity fields.

As stated by authors, maps of geostrophic velocity error do not exist yet. They used two complementary approaches to assess the robustness of the Lagrangian technique to the presence of these errors. I think that they did all their best to follow reviewer's recommendation.

I have an additional question about the trends mentioned in the manuscript. In particular I would like the authors considering to perform a significance test of the estimated trends that is missing in the actual version.

In summary, concerning the satellite altimetry observations, the authors have made a great effort to follow reviewer's recommendation. Before publishing, there is a need to better quantify the quality of the data set and some numbers about accuracy need to be provided in the main manuscript (a

synthesis from the literature and comparison showed in the supplementary section). In the present for, there are only generic sentences. More quantitative results need to be added.

Finally, about SST and ocean color data sets, the authors use standard products available from providers. As for altimetry, the authors have to comment the quality and accuracy of SST data in the area of investigation. I am not here to ask more analysis, but a synthesis from the bibliography about the confidence in using these data sets is particularly necessary for the broad readership of this journal.

I am available to have a check of the revised version.

Minor comment

Please provide reference of Cipollini et al. 2014.

We would first want to thank the Editor for taking charge of this review process.

We have done our very best to answer every concern.

We have done substantial changes in the main manuscript but also in the Supplementary Material as well in the hope that the improvements will satisfy your requirements.

The major modifications are:

- The supplementary data sections 1.1 and 1.2 about altimetry, SST and ocean color satellite data in the SI have been brought in the material and methods information in the main text, but also updated and improved following reviewers #4 and #5 recommendations. We have significantly enhanced the description of the data sets for more transparency and reproducibility. More importantly, we have put more weight (using quantitative results) on the altimetric data set's strengths which have been demonstrated (validated) in this work or in other recent studies.
- Answering reviewer #4 concerns, we have better linked the main manuscript with the Supplementary Result especially about the statistical analysis.
- Also, in the main text, we added in the introductory section, considerations about other environmental factors that could influence Ehux growth. It better connects with the discussion later on (reviewer #4).
- We have removed the term 'thermal barrier' in agreement with reviewer #2. We now consider 4degC as "the lowest temperature required for sufficient growth to allow bloom formation". This is a long definition but we believe this is the most appropriate. We have completely re-written and improved the first paragraph of section 2.5 in the SI and the main text in accordance with reviewer #2 recommendations.
- Some referencing issues have been corrected
- We added a *p*-value map for velocity trends and we draw mooring positions on Figure. S1.
- Some small other modifications have been done in the writing or in figures and can be found in the tracked-change manuscript or in the list at the end of this document.

Please, find below our point-by-point answers.

Reviewers' comments:

Reviewer #2 (Remarks to the Author):

I review for a second time the manuscript by Oziel and colleagues. The revisions produce in this second manuscript appear to be extremely well done. The manuscript is easy to read, and I do not have any problem with it. The authors have answered well to all my remark (except one, see below). I read with attention how they answered the question of Review 1: apparently the reviewer was not as positive I was on the 1st manuscript, most of his/her remarks are related to the consideration of older work. The author in the second version produced a noticeable effort in taking carefully his/her advice. I shared the same feeling about 'thermal barrier,' and on this point they do not answer completely, as they just explain in more detail what they mean by thermal barrier. I will come back to that later. For the rest they answer positively to all questions of Rev. I.

There is only one minor point for which I am worrying on the meaning of the 'thermal barrier.' They write:

“The 4 °C surface isotherm (Fig. 4) is considered as a thermal barrier for EHux because it represents the lowest temperature required for their growth (Supplementary Result 2.5). However, the Ehux blooms do not seem to follow this thermal barrier in summer and appear to be constrained by another factor. For example, the north-eastward expansion of the winter 4 °C isotherm, which delimits the inoculum area in our Lagrangian method, could contribute to reduce the distance between the Ehux winter location and the Arctic domain.”

In the supplementary material is said: 'Despite the fact it is exceptionally apt at surviving in cold conditions ($T = 0\text{ °C}$), most studies (ref. 35–37) agree that its temperature niche is defined by temperatures above 3 °C'

I checked Ref.35 (Balch 2005) and Ref.34 (Paasche, 2001). Because I think there is a shift of the references 34 in the list is referred as 35 in the text.

Note: true, we have corrected that referencing issue in the SI.

Paasche mentions a limit at 1 °C and Balch does not speak of 3 °C. I did not find Ref.36 (Vu, B.Le) because the reference is not complete.

There is no agreement on that growth limit.

Again later, it is said 'the studies (ref.40,41) compiling the most extensive field collection of Ehux data in which 4 °C was the coldest temperature observed that was associated with non-zero growth rates of this phytoplankton.'

Ref 39 (=40 in the text) mention a limit at 4 °C, but from culture not in situ. Growth rate was given from literature ('E. huxleyi growth rate data were obtained from culture experiments detailed in the

primary literature'). It is very difficult to grow E.hux or any other organism at its limits of adaptation. The collection of strains, in Ref. 39 are in the range of 5 °C-25 °C. Therefore it is not surprising that the only attempt at 4 °C was not successful.

I did not find any reference on low temperature in Ref 40 (=41 in the text)

Note: Thank you, there is a confusion here. We meant Iglesias-Rodriguez et al. (2002) and not Balch (2004). This has been corrected. We have investigated the correct reference for the PhD from Briac Le Vu (2005), but given that the PhD thesis is in French and the reference is not essential for the demonstration, we preferred to remove the citation.

It is not possible that there is a lower growth limit at 3 or 4 °C: if below 3 °C *E.huxleyi* would not grow, how could we find this species at 0 °C (I collected it at that temperature in several occasions many studies show that) ? In conclusion, if *Ehux* is found at lower temperature than 3 °C it has to grow below 3 °C !

This is a minor point raised by Reviewer 1 and myself. It is minor because it does not change the manuscript. It is just a problem of semantic. 'Thermal barrier' and 'Ehux minimal temperature for growth' could be changed by something like 'Ehux minimal temperature for regular growth.'

Beside this small problem, I found the paper extremely interesting well written, and highly important for our understanding of the Arctic changing ecosystems.

We would first like to warmly thank the reviewer #2 for taking the time to review this manuscript again and also for providing such constructive comments. Thank you for raising this last point for which we agree that it needs clarification. We have changed in the main text "Ehux minimal temperature for growth" to "[Ehux] lowest temperature required for sufficient growth to allow bloom formation" as indicated. The sentences become:

"The 4°C surface isotherm (Fig. 4) is considered for Ehux as the lowest temperature required for sufficient growth to allow bloom formation (Supplementary Result 2.5). However, the Ehux blooms do not seem to follow this isotherm in summer and appear to be constrained by another factor."

First, we agree that we cannot prove that *Ehux* does not grow below 4°C. The only thing that we could eventually claim is that they are unable to form significant blooms below this temperature since almost no PIC is found below 4°C (Signorini and McLain, 2009). We hypothesize that this would be due to very low growth rates at low temperatures as evidenced by culture experiments. But we also agree that results from cultures may differ from the *in situ* reality.

- 1) Fielding et al. (2013) which is, to our knowledge, one of the largest collections of laboratory data at present time, specifically studied growth rate dependence of temperature of *Ehux*. They showed 4 experiments below 6°C. Two of them with null growth rates, and the other two with very low ones (~0.1 d⁻¹). The lowest temperature with non-zero growth rate is 4°C. A recently published study (Wang et al, 2019, <https://doi.org/10.5194/bg-16-4393-2019>), using a model based on lab culture, demonstrated that the minimum temperature for *Ehux* non-zero growth rate is about 6°C (Wang et al. 2019):

Figure 1. Thermal performance curve showing cell-specific growth rates (d^{-1}) of *Emiliana huxleyi* CCMP371 across a temperature range from 8.5 to 28.6°. Symbols represent means, and error bars are the standard deviations of three replicates at each temperature, but in many cases the errors bars are smaller than the symbols.

Figure 8. Thermal performance curves based on specific growth rates (d^{-1}) of *Emiliana huxleyi*, including our experimentally determined constant condition temperature performance curve (black symbols and solid line) and a predicted fluctuating condition temperature performance curve (dashed line) according to the model of Bernhardt et al. (2018). Measured growth rates from the two low- and high-temperature experiments are shown for constant thermal conditions (red symbols), 1 d (green symbols) and 2 d (blue symbols) variation treatments.

- 2) Remote sensing studies cannot tell if phytoplankton is growing with a biomass indicator such as the PIC. But the PIC can tell us if there is a bloom (i.e. biomass accumulation = when growth overestimates losses). Remote sensing studies have consecutively defined 3, 5 and 6°C (Iglesias-Rodriguez et al. 2002; Signorini and McLain, 2009; Neukermans et al. 2018) as the lowest temperature for the detection of an EHux bloom from space.

To conclude, we decided to choose a threshold of 4 °C on the base of Signorini and McLain (2009) who concluded that: “almost no PIC was detected below 4 °C” and that it corresponds to the minimum temperature for non-zero growth rate (Fielding et al. 2013) consistently with the new definition: “the lowest temperature required for sufficient growth to allow bloom formation” of Ehux. Therefore, following your recommendation, we have also thoroughly rewritten the first paragraph of Supplementary Material 2.5 to clarify:

“The temperature range for Ehux maximum growth rate is estimated to be around 21°C²³ with lower and upper boundaries of 10 and 26°C²⁴. Below 10°C, the growth rate is not optimal anymore and is expected to decrease. The current largest collection of Ehux growth rates (laboratory data) demonstrated that the lowest temperature with a positive growth rate is at 4°C^{25,26}. More recently, a model implemented from laboratory data indicated that the growth rate is becoming close to zero around 6°C²⁷. However, laboratory studies may underestimate actual Ehux growth rates at low temperatures. In the vicinity of the EAC, remote sensing studies observed different minimum temperature thresholds for the detection of Ehux blooms. This threshold was initially found to be 3°C²⁸, then updated²⁹ to 5°C and more recently defined as 6°C¹. This temperature range is typical of Atlantic Waters². On exceptional occasions, Ehux are even apt at surviving in polar conditions³⁰ (T ~ 0°C) but with expected low to very low growth rates. In the absence of consensus, and because almost no PIC is detected below 4°C in the EAC²⁹, we decided to choose for this study 4°C as threshold for the minimum temperature required for a “regular” growth that could allow the formation of a bloom. Below this limit, we consider that the Ehux development is not sustainable with either lower growth rate than losses or close to zero growth rate.”

Reviewer #4 (Remarks to the Author):

This paper claims that an increase in the intensity of currents in the North Atlantic can explain the shift towards the Arctic of the coccolithophore *Emiliana huxleyi* (Ehux). The authors use as input data a time series of 25 years of standard satellite altimetry to estimate surface currents, two satellite-derived products (sea surface temperature) and particulate inorganic carbon concentrations derived from satellite ocean color as a proxy for coccolithophore biomass. In addition, a lagrangian code is run to estimate particle trajectories.

The results are to my knowledge novel and could be of interest to others in the community as there is an attempt to show the potential impact of surface currents on a marine species. The paper is well written and follows a logic flow but unfortunately it is far from being convincing specially for a high-impact journal such as Nature Communications.

We thank reviewer #4 for the thorough reading of our paper.

I present below my general comments:

- 1) Results are not well supported by the data because they do not demonstrate the real relation between currents and coccolithophore. What they are showing is just a correlation but not a demonstration that the primary driver of the Ehux dynamics are surface currents. I am not an expert in this area in Ehux dynamics but I suspect the problem is highly more complex and the approach followed in this paper is oversimplified.

1. We believe we did demonstrate the link between ocean currents and coccolithophores and that results are supported by data. In fact, the demonstration between ocean currents and Ehux poleward expansion is not based on a correlation analysis. A whole paragraph (first one of the section “Increasing NAW currents control the spatial distribution and poleward expansion of Ehux”) is dedicated to show that the spatio-temporal bloom dynamic is

realistically reproduced from ocean currents only (see Exp.1). This is a strong result which was also supported by a quantitative statistical analysis in Supplementary Material 2.7 (match-ups between remotely sensed Particulate Inorganic Carbon and virtual particles). Once this link between currents and phytoplankton dynamics was demonstrated, we investigated the temporal trends. To do this, we used this time correlations to discover whether it was temperature or currents that drove the inter-annual variability of PIC distributions. The long-term evolution was assessed using trend statistics. However, we recognize that the way we have written this paragraph could lead to confusion. To clarify the text, we first, in the introduction, replaced the term “advection” by “ocean currents”:

*“Here we investigate how **ocean currents control** the spatial dynamics of a specific coccolithophore bloom-forming species, *Emiliana huxleyi* (from hereafter referred to as *Ehux*).”*

In the section “Increasing NAW surface currents control the spatial distribution and poleward expansion of *Ehux*.”, we also admit the first sentence introducing the demonstration did not express clearly the purpose of the experiment:

Former sentence: *“The hypothesis that northward advection of *Ehux* explains the summer distribution of *Ehux* in the BS was tested using a Lagrangian experiment (EXPI).”*

It is now replaced by:

New sentence: *“The **first hypothesis that ocean currents control** the summer **spatial** distribution of *Ehux* in the BS was tested using a Lagrangian experiment (EXPI).”*

At the end of the demonstration, we insist:

*“The year-to-year robustness of the matchup between virtual particles and PIC clearly supports the fact that the NAW surface currents shape the location and extent of *Ehux* blooms in the EAC”.*

To conclude, this study’s objective is not to say that ocean currents are driving *Ehux* development/growth, but that ocean currents are shaping their spatial distribution and poleward intrusions. The result has been demonstrated in the experiment #1.

We hope that these changes are satisfactory.

2. Apart from the changes that we’ve made regarding the relation between advection and *Ehux* dynamics, we agree that *Ehux* dynamics are affected by other numerous environmental factors such as light, mixing, nutrients, pH...

But because we wanted the manuscript to highlight one physical forcing in particular (ocean currents), we did not focus our demonstration on other factors in the results but we did consider them in the discussion (light, nutrients, mixing, grazing). At the end of the first paragraph of the last section “**A newly “Atlantified” and retreating marine Arctic ecosystem**”, we say:

“Ehux seems to be adapted to the low light, low nutrient, oligotrophic and highly stratified conditions of the NAW in summer³² such that its expansion, growth and blooming in the Arctic Ocean will be limited at some point by those constraints (bottom-up) but also by the grazing pressure (top-down).”

We further suggest:

“Despite its adaptation to low light conditions, Ehux still requires sufficient light levels to sustain the energy-demanding calcification of its coccoliths, which are met in highly stratified oceans where it maintains in the surface layer⁵⁶”

Finally, we speculate about nutrients:

“The concomitant decrease concentration in silicates in NAW⁶³ may also contribute to the success of non-silicifying and small phytoplankton such as Ehux⁶⁴.”

However, following your advice, we did a substantial addition in the first section of the main article to better consider environmental factors from the beginning of the article:

*“Here we investigate how advection controls the spatial dynamics of a specific coccolithophore bloom-forming species, *Emiliana huxleyi* (from hereafter referred to as *Ehux*). ***Ehux is typically associated with the surface layer of the temperate NAW in summer, typically in a post-spring bloom context characterized by low nutrients, low light and strong stratification³²⁻³⁴***”*

The references 32, 33 and 34 are respectively:

Balch 2004

Paaschee 2001

Gafar and Schulz 2018

They are among the most complete studies, to our knowledge, about Ehux’ ecological niche. The format of the journal does not allow us to be completely exhaustive, but the reader can now more easily refer to those articles. We believe this additional information will better connect with the last discussion section entitled “A newly “Atlantified” and retreating marine Arctic ecosystem”.

Again, we would like to repeat that this study’s objective is not to say that ocean currents are driving Ehux growth, but that ocean currents are shaping their spatial distribution and poleward intrusions.

- 2) I have serious concerns on the appropriateness and validity of the data used and the statistical analysis performed. In particular, what is the robustness of the trends shown in figure 5? It is necessary to perform an error analysis as a trend without an associated error is meaningless.

We acknowledge this is an important point but the trends have been already derived together with a suite of statistical indices. Indeed, we provided in the last manuscript (see former

Supplementary Material) Pearson's correlation coefficients, slope and offset together with their respective 95% confidence intervals. We also derived p -values. The statistics were shown in a table in section 2.9 (now 2.8) of the Supplementary Material. Leading edge trends over the 1998-2016 period all showed p -values < 0.05 , evidencing significance levels $> 95\%$. But we are aware that this information does not appear anywhere in the main text and lies at page 21 in the Supplementary Material. Therefore, we have now referred to it in the last section of the main text. We modified:

“This significant increasing trend in current velocities (see Supplementary Result 2.8) is also revealed by the greater distance covered by the virtual particles reaching the BS, which increased by 110 km on average since 1993 (EXP2, Supplementary Result 2.9).”

We would like to add that during the last round of review, we have addressed the issue of error/uncertainty associated with satellite-derived geostrophic velocities by performing a “monte-carlo” analysis. This method has already been employed to test the robustness of the Lagrangian analysis. We repeated 1000 times the same calculation adding each time a random diffusion component. You can find a detailed description in Supplementary Result 2.4. We believe this is a powerful means to show uncertainty, and we thereby showed the trend was robust to the addition of noise. However, in response to the reviewer's concern along with a comment from reviewer #5, we have now added a p -value map in the Supplementary Information (former figure Figure. S9, section 2.3) about the trend in geostrophic velocities in the EAC.

3) Even if altimetry is a major tool for studying ocean circulation and has revolutionized our understanding of mesoscale variability, its reliability in an area so close to the North pole and to the coast as in this study is doubtful. Due to the specific orbit inclination of TOPEX/POSEIDON and the Jason series (the reference high accuracy altimeter missions), at latitudes higher than 66°N , the amount of data is very limited as it only comes from other missions such as ERS, Envisat, with lower accuracy and with a temporal repetitivity of 35 days, even if with a higher spatial resolution. Regarding the coastal band, it is well known that standard altimeter grided products in the coastal zone are adversely affected by land contamination with degraded range and geophysical corrections (see for example Vignudelli et al. 2011). A comparison with in situ data from currentmeters (for example) is required in order to show the potential validity of altimetry in this area.

REFERENCE

Vignudelli, S., Kostianoy, A. G., Cipollini, P., and Benveniste, J. (eds). (2011). Coastal Altimetry. Berlin: Springer-Verlag, 578. doi: 10.1007/978-3-642-12796-0

We entirely agree with the reviewer that the data availability at high latitudes is lower due to the absence of some satellites North of 66°N (T/P, Jasons...) and that this issue deserves a dedicated validation. But such work, such as validation toward current meters, has already been undertaken in this manuscript (Supplementary Result 2.1). We have now referred to it more clearly in the material and method in the main text.

First, it is important to note that the multi-mission product produced by TAC and distributed by CMEMS has been subject to a large number of corrections and improvements in recent years and rely on different satellites such as ERS-1, ERS-2, Saral/Altika, GFO, Envisat and recently CryoSat-2 and Sentinel-3A (see also answer to reviewer #5).

Furthermore, the dataset has been previously validated (SLA and geostrophic currents) with the most complete collection of drifters and tide gauges data in the EAC (see Volkov et al., 2012, 2013; Vignudelli et al., 2011). The Figure below show the positions of the drifters (i.e. where the satellite altimetry has been validated) and exactly corresponds the Atlantic domain considered in our study.

Figure 4. (a) The trajectories of surface drifters present in the region from January 1993 to December 2010. (b) The number of drifter records in $1^\circ \times 1^\circ$ bins.

The authors compared the velocities from drifters with those from altimetry (corrected from Ekman) and found good RMS differences (7-15 cm/s) typical of those found at lower latitudes.

About data coverage, the Figure from Volkov et al 2012 below shows an increase in mapping error fields above 66degN which is due to a lower number of satellite tracks. However, the errors still remain moderate with a good data coverage everywhere except in the ice-covered areas (northeast Barents Sea, Kara Sea, Greenland Sea) which are out of the scope of this paper and carefully avoided.

Figure 2. (a) Mapping error field (percentage of the signal variance) on 28 August 2008 given by the optimal interpolation of Jason-2/OSTM and Envisat measurements. (b) Portion (%) of missing weekly altimetry records.

The authors concluded:

“In summary, in this paper we have provided a rather strong validation of a modern satellite altimetry gridded product in the Nordic seas. This means that this product can be successfully used to study the variability of sea level and surface circulation in the region. However, some limitations have to be taken into account. We have shown that in some coastal areas of the Kara Sea the agreement between the altimetry and tide gauge data is relatively poor.”

In the Norwegian Sea, Raj et al. (2018) again validated the altimetry dataset and concluded that it was especially appropriate to monitor the long term northward Atlantic inflow. They found better RMS differences (between drifters and satellites) than Volkov et al. 2012 (0.7-8 cm/s)

In addition, in the supplementary material of this study, we insist we have already investigated satellite-derived geostrophic velocities toward 5 *in situ* current meters at the Barents Sea opening, finding fairly good Pearson’s correlation coefficients ($r = 0.43-0.7$) typical for a validation between *in situ* current meters and gridded altimetric product.

See for example, in the Table 4 of the following article, Pearson’s coefficients typical for the well known and validated Malvinas Current:

Ferrari, R., Artana, C., Saraceno, M., Piola, A. R., & Provost, C. (2017). Satellite altimetry and current-meter velocities in the Malvinas Current at 41degS: Comparisons and modes of variations. *Journal of Geophysical Research: Oceans*, 122, 9572–9590. <https://doi.org/10.1002/2017JC013340>

It is also true that near the coast discrepancies can be larger but in the EAC area, RMS estimations of geostrophic velocities were low (see below answer to reviewer #5) and SLA was minorly affected by coastal contamination in the Atlantic domain of the Norwegian and Barents Seas (Volkov et al. 2012). This is true that more discrepancies were found in the Kara Sea and the White Sea which are again carefully avoided in this study. In chapter 15 of the book ‘Coastal Altimetry’ that the reviewer quotes, the authors evidenced very good correlations between tide gauges and satellite altimetry (GFO, ERS-1/2, Envisat) for the coastal Barents Sea with very high correlations coefficients ($r \sim 0.99$, see below). See the attached figure from their article.

Fig.15.8 Time variability of Vardo TG and ERS-2 & Envisat SSHA (a) and scatter plot of TG data versus altimetry data (b)

They concluded: “According to the results obtained, satellite altimetry SSHA data are in good consistency with TG SSHA.” Note that TG = Tide Gauge

“Correlations are extremely high (0.99) when ERS-1/2 and Envisat data are used for the comparison with Vardo and Honningsvåg TGs for which there are long and high quality time series (Lebedev et al. 2008).”

Note that Vardø and Honningsvåg are the coastal tide gauges which are at the northernmost position of Norway and the closest to our study area (and virtual particles). The correlations decrease toward the White Sea which is a very coastal area outside of our domain of interest.

Finally, using only ERS-1/2, Envisat and GFO, they concluded (section 15.5):

“Spatial and temporal coverage of the Barents and White seas by satellite altimetry missions offers the possibility of continuous high-precision monitoring of sea level, ice cover, and sea surface wind speed [...] The results obtained seem to be promising in application of satellite altimetry data on SLA and wind speed for monitoring a number of environmental parameters of the Barents and White seas.”

Furthermore, the dataset used in this study benefited from the processing from TAC and additional satellites (CryoSat-2, Sentinel-3A). The SL-TAC paid particular attention to synchronize, homogenize, cross-calibrate, correct large-scale biases and cross-validate all missions.

Considering the reviewer’s concerns, and those from reviewer #5, we have thoroughly modified and improved the data and material in the main text to (1) better present the data set, (2) explain the strengths and caveats of the data set, by using the ‘state-of-the-art’ of satellite altimetry validation studies in the area we just described, and (3) expressing quantitatively our validation with in situ current meters and previous ones with drifters. Please, find the completely new data section of in the new version of main text or below:

“Satellite-derived products.

1- Altimetry.

The satellite altimetry product was generated by the Sea Level Thematic Assembly Center (SL-TAC) and distributed by the Copernicus Marine and Environment Monitoring Service (CMEMS, <http://marine.copernicus.eu/>). This is a daily gridded global product in delayed-time (re-analysis product ID: SEALEVEL_GLO_PHY_L4_REP_OBSERVATIONS_008_047) with a spatial resolution of 1/4° available from 1993 to present time. A 1/4° Mercator projection grid roughly corresponds to a resolution of 27.8 km in latitude x 9.5 km in longitude at 70°N. This product is generated using Sentinel 3A, Jason 3, HY2A, Saral/AltiKa, Cryosat 2, OSTM/Jason 2, Jason-1, Topex/Poseidon, Envisat, GFO, ERS-1/2. The SL-TAC produced this data set with a special care for synchronization, homogenization, cross-calibration, correction of large-scale biases and cross-validation. Before merging missions by optimal interpolation, the SL-TAC filtered the data set to remove residual noise. The reader is referred to the product documentation for more details. The satellite altimetry product provides Sea Level Anomaly (SLA) that estimates the variation of the Sea Surface height around a vertical reference (Mean Sea Surface). The product also provides Absolute Dynamic Topography ($ADT = SLA + \text{Mean Dynamic Topography MDT}$), geostrophic velocity anomalies and absolute geostrophic velocities (derived respectively from SLA and ADT). The MDT employed by the TAC was the MDT-CNES-CLS13 produced by the Collecte Localisation Satellites (CLS) Space Oceanography division and distributed separately by Aviso (<http://www.aviso.altimetry.fr>).

SLA has been successfully used over the last 25 years to study the variability of sea level and surface geostrophic circulation at different spatial and temporal scales. However, most of the validation studies have focused on the areas below the polar circle ($< 66^\circ\text{N}$) such as in the Norwegian Sea where derived geostrophic currents were successfully evaluated against in situ current meters⁷⁰. The use of altimetry in Arctic regions has been avoided in the past due to (1) persistent sea ice cover and (2) because some radar altimeters (TOPEX/POSEIDON, Jason-1, OSTM/Jason-2 and Jason-3 missions) do not acquire measurements above the polar circle. First, sea ice was not an issue for this study because virtual particles were only evolving in the year-round ice-free Atlantic domain. For the trend

and EOF calculation, data associated with sea ice were systematically precluded (see below). Second, measurements above 66°N can rely on several other satellites: ERS-1 and -2, Envisat, SARAL/AltiKa, GFO (<72°N), HY-2A, ICESat, Cryosat-2, Sentinel-3. A comprehensive validation of the gridded product of geostrophic velocities only from ERS-1/2 and Envisat satellites using surface drifters and tide gauges in the whole EAC, with a particular focus on the Barents Sea, was recently provided^{71,72}. The authors demonstrated that RMS differences between the drifter (corrected for Ekman currents) and altimetry velocities range within 7-15 cm s⁻¹ which is comparable to previous estimates at lower latitudes. More importantly, they also concluded that the data set provides sufficient spatial and temporal coverage to consistently resolve the synoptic and large mesoscale variability. Moreover, the product was recently found appropriate to monitor the long term northward Atlantic inflow in the Norwegian Sea⁴² with even lower RMS differences between drifters and satellites velocities (0.7-8 cm s⁻¹).

The dataset used in this study was enhanced from additional satellites (i.e. CryoSat-2 and Sentinel-3A) that offer a better data coverage at high latitudes for recent years (since 2011). Surface currents inferred from altimetry were found to reflect the variability of the deeper layers in the slope current in the Nordic Seas⁴². For this study, the derived surface geostrophic velocities were evaluated again toward five long-term in situ current meters in the southwestern Barents Sea and showed reasonably good Pearson's correlation coefficients ($r = 0.43-0.71$, Supplementary Result 2.1). In addition, by evaluating the effect of steric and mass related effect on the SLA, we demonstrated that the increasing long-term trend was driven by change in currents (Supplementary Result 2.2).

”

Reviewer #5 (Remarks to the Author):

The editor asked me to assess if authors addressed properly technical points related to usage of altimeter data that were raised by a reviewer who was unable to help at this round.

In this paper, the authors use satellite-derived altimetry observations to study the North Atlantic current surface velocities in the Arctic Ocean over the last 24 years. Their interest is at the long-term temporal scales (interannual and trends).

During the last decade, standard altimeter data underwent to several reprocessing campaigns with the objective of homogeneous different missions and inclusion of new/improved corrections, better inter-calibration, updated vertical references in order to estimate sea level anomalies. In parallel, some scientists made validation of these products in specific regions to assess their accuracy for exploitation in oceanographic studies. The actual multi-mission products available via the Copernicus Marine Services (<http://marine.copernicus.eu/>) are benefiting of these enhancements and represent the state-of-the-art to study global and regional ocean circulations.

Therefore, this paper is one excellent example of fully exploitation of one of these products. It is also important to highlight that there are ongoing efforts in extending the satellite-based sea level record in challenging domains (coastal zone, inland waters and sea ice regions). Moreover, the European Space Agency Climate Change Initiative (CCI) project on “Sea Level” has reprocessed altimeter data to provide satellite-based sea level products to study climate-scale variations of sea level globally and in the coastal zone. The global product (available via ESA) provides monthly means of Mean Sea Level Anomaly @¼ degree during

1993-2015 using TOPEX/Poseidon, Jason-1, Jason-2, ERS-1, ERS-2, GeoSat Follow-On (GFO), Envisat, SARAL/AltiKa and CryoSat-2 with special care of editing, cross-calibration, homogeneous corrections, removal of global and regional biases, homogenization of long-spatial-scale errors, monthly optimal interpolation gridding. Major details at Legeais, J.-F., Ablain, M., Zawadzki, L., Zuo, H., Johannessen, J. A., Scharffenberg, M. G., Fenoglio-Marc, L., Fernandes, M. J., Andersen, O. B., Rudenko, S., Cipollini, P., Quartly, G. D., Passaro, M., Cazenave, A., and Benveniste, J.: An improved and homogeneous altimeter sea level record from the ESA Climate Change Initiative, *Earth Syst. Sci. Data*, 10, 281–301, <https://doi.org/10.5194/essd-10-281-2018>, 2018.

The authors in this manuscript use the satellite altimetry product generated by the Sea Level Thematic Assembly Centre (TAC) and distributed by the Copernicus Marine and Environment Monitoring Service (CMEMS). This is a gridded product with a spatial resolution @ $\frac{1}{4}$ degree with 1-day temporal resolution during 1993 to 2016. It is generated using Sentinel 3A, Jason 3, HY2A, Saral/AltiKa, Cryosat 2, OSTM/Jason 2, Jason-1, Topex/Poseidon, Envisat, GFO, ERS-1/2). The satellite altimetry system provides Sea Level Anomaly (SLA) that takes into account the variation of the Sea Surface height (orbit minus range corrected) around a vertical reference (Mean Sea Surface).

SLA from CCI and TAC products are not exactly the same as the processing chains are different. The authors use the SLA TAC because their interest is not in sea level but in ocean velocities. Therefore, they need daily data sets to use in synergy with the Mean Dynamic Topography (MDT) provided by the CNES-CLS13 dataset produced by the Collecte Localisation Satellites (CLS) Space Oceanography division and distributed by Aviso to compute the Absolute Dynamic Topography (ADT=SLA +MDT) from which to derive the velocities.

We thank the reviewer for these insightful comments. We entirely agree we could not have used CCI for the needs of the Lagrangian experiments (daily velocity fields are needed). However, we have now referred to this dataset as an alternative for other studies. We also added that using already available products that provide modelled Ekman velocities (and also Stokes drifts), we could improve the precision of the satellite ocean currents estimations (we have referred to Liu et al. 2012 following your advice). We were not aware of this data set at the time we started the analysis (or perhaps it did not exist yet) but we clearly say that this will be the next improvement to consider for further studies.

We added these informations at the end of section 2.1 in the SI:

“We note that recently, CMEMS is distributing a dataset produced by the Centre National d’Études Spatiales (CNES) and the Collecte Localisation Satellites (CLS) in the framework of the Globcurrent ESA project (<http://www.globcurrent.org/>) that delivers satellite-derived geostrophic velocities together with modelled Ekman transports and Stokes drift. A future inclusion of the Ekman and Stokes components could further improve the estimation of ocean currents and reduce errors¹¹ and should be considered for further studies. Alternatively, the European Space Agency Climate Change Initiative (CCI) project on “Sea Level” has reprocessed altimeter data to provide satellite-based sea level products to study climate-scale variations of sea level globally¹² and in the coastal zone¹³. The global product (available via ESA) provides monthly means of Mean Sea Level Anomaly $1/4^\circ$ during the 1993-2015 period using TOPEX/Poseidon, Jason-1, Jason-2, ERS-1, ERS-2, GeoSat Follow-On (GFO), Envisat, SARAL/AltiKa and CryoSat-2 with special care of editing, cross-calibration,

homogeneous corrections, removal of global and regional biases, homogenization of long-spatial-scale errors, monthly optimal interpolation gridding. The need for daily temporal resolution for the Lagrangian experiments hindered us to use this data set.”

Having said that, the authors estimate an altimetry-based velocity field for the time period 1993 to 2016. The main point raised by the previous reviewer concerned the assessment of this field in terms of accuracy (the so-called validation using independent data sets) over the area of investigation. This is an important aspect of the analysis in order to support the acceptance of the scientific findings (i.e. that the observed increase of the water volume during the 2000s is error free and volume changes are not much larger than the expected error bars of the measurements).

The authors agree that the computed velocity fields need to be supported with more explanations and quantifications about their accuracy.

In the manuscript, section “Material and Methods”, the authors briefly describe the altimetry data sets and following reviewer’s comment a paragraph about validation is added with a pointing to supplementary for more details (section “1.1. Details on the altimetry dataset).

I think the paragraphs need some rewording to avoid confusion in the reader. In the manuscript, the authors have just to specify the products they use, i.e. SLA TAC, MDT with source, spatial and temporal resolution. Then the authors explain the method ($ADT=SLA+DT$) and derivation of geostrophic velocities. For example the sentence “excluding pixels with error estimates (percentage of the signal variance) higher than 50” that is a technical detail has to be moved in the supplementary part, where I would remove “The Sea Level data are corrected for instrumental errors, inverted barometer, tides and tropospheric effects” because it is too generic (a table with specific sources would be necessary or a product reference) and incomplete (e.g., we need to correct also for ionosphere, SSB, DAC). DAC (Dynamic atmospheric correction) that includes wind effects is necessary to reduce the aliasing of the high-frequency sea level variability in altimetry data, especially in coastal regions. I also suggest to specify “The Sea Level Thematic Assembly Centre (TAC)” when CMEMS is mentioned and explain the exact product that is used (as there are different products, e.g., NRT, reprocessed, global, regional specific).

Thank you very much. Thanks to your advice, we were able to completely re-design the material and method in the main text in order (1) to give a more accurate description of the data set; (2) to better illustrate the ‘state-of-the-art’ bibliography about satellite altimetry in the area; (3) but also to give quantitative results about our own validation and previous ones.

This transformation allowed us to transfer and summarize all the information contained in supplementary sections 1.1 and 1.2 directly in the main text and to bring a large number of improvements following your recommendations.

- Therefore, former sections 1.1 and 1.2 in the SI about altimetry and ocean color satellite data do not exist anymore (moved and merged in the Material and Methods in the main text)
- We have referenced the product ID so that the reader can find all necessary information about the dataset on the CMEMS web site.

- We have listed the general features of post-processing done by SL-TAC in order to ensure a good quality of the data set (i.e. synchronization, homogenization, cross-calibration, correction of large-scale biases and cross-validation). Of course, we cannot be exhaustive concerning the numerous specific corrections which are made (e.g. wind aliasing correction, invert barometer, etc) and would necessitate tens of pages but the reader can now refer directly to the product documentation.
- In a long last paragraph in the data section of the main text, we now discuss the limits and strengths of the data set by summarizing the validations studies in EAC area. We also expressed quantitatively the results from our validation with current meters at BSO together with the previous validation studies (Pearson's correlation coefficients, RMS differences with drifter velocities)
- We discuss why the sea ice is not an issue for this study because we only focus on the Atlantic domain which is permanently ice free. For the trend calculation, we indicate that we carefully removed data contaminated with sea ice.
- Concerning data coverage, we quote existing studies that evidence a sufficient data coverage for large mesoscale and synoptic monitoring using only ERS 1-2 and Envisat. Following your advice, we insisted the data set is also benefiting from CryoSat-2 and Sentinel-3A.
- The new organization required the modification of the EOF analysis method section (main text) in which we summarized how we derived monthly fields.

The only thing that we did not manage to do is to move the sentence “excluding pixels with error estimates (percentage of the signal variance) higher than 50” in the SI. The corresponding section do not exist anymore. The sentence is now in the section “***Empirical Orthogonal Function (EOF) analysis of SLA and trends in surface geostrophic velocities.***” of the method section in the main article. We apologize for this but we believe this new organization of the manuscript is clearer for the reader.

Please find in the tracked change main manuscript the new data and method sections which has been thoroughly improved following all your recommendations. Also find below detailed point-by-point answers to your concerns. We hope that those improvements will satisfy you.

Then the authors have to convince the reader that the dataset they use is of quality and accurate. Concerning the quality, as highlighted by Volkov (2012) there is a problem of altimetry coverage. The advantage of the SLA TAC (compared to the data set used by Volkov) is the usage of CryoSat-2 that envelopes the Arctic region up to 220 km from the pole, while all the other missions have inclination allowing to cover the Arctic Ocean not closer than 950 km (81.5 deg North). Moreover, there is the problem of the seasonal presence of sea ice for which the altimetry data in the study region have gaps (see Volkov 2012, Figure 2). I think the authors do not provide any clarification about the new product with reference to satellite coverage (CryoSat-2 was launched in 2009 so what about before?) and presence of sea ice that is and indicator of the quality of data used.

We agree with the reviewer that these issues deserve clarification.

1) The presence of sea ice is an important caveat of altimetric data set. We now discussed why this is not an issue for our study in a second paragraph of the material and method (can be found below).

We clarify that data associated with the presence of sea ice is not an issue for the Lagrangian trajectories because virtual particles are exclusively evolving in the Atlantic year-round ice-free domain where the data coverage is persistently close to 100% in reference of the Figure 2 from Volkov et al. 2012 (shown above for reviewer #4). However, it could be an issue for the trend or the EOF calculation in ice covered areas. In this case, we made sure to avoid areas with sea ice contaminations. For example, any data point associated with >15% sea ice concentration was removed. And no trend was derived for time series with less than half of data points. See the following new section in the Data and Methods:

“Empirical Orthogonal Function (EOF) analysis of SLA and trends in surface geostrophic velocities.

EOF and trends analysis were performed between 1993 and 2016, with monthly composite maps of non-seasonal SLA and surface absolute geostrophic velocities. Monthly composite maps were derived excluding pixels with error estimates (percentage of the SLA signal variance) higher than 50% or sea ice concentrations higher than 15%. The seasonal cycle, known to be highly variable for SLA and the velocity fields, was removed because the analysis focuses on the inter-annual and decadal time scale. The maximum amplitude in SLA generally occurs in fall⁸⁵, whereas the velocity fields are maximum during the winter⁸⁶. More details on the EOF analysis are available in Supplementary Result 2.3. Trends in surface absolute geostrophic velocities were only derived for pixels associated with more than 50% of data points.”

Note that 30-days composite maps were constantly close to 100% data coverage with the Atlantic domain (~ -8 – 60 degree LON 50 – 76 degree LAT) despite the masking of data associated with sea ice concentration (15%) or large variance in the SLA fields (>50%). For the trend analysis, pixels associated with time series with less than 50% data points were masked as well.

We added at the end of Figure 3 (and Figure S8) caption:

“Areas covered by sea ice (sea ice concentration > 15%) or with insufficient data coverage for the trend (< 50 %) are in dark gray.”

For the EOF analysis, masked data were replaced by zeros because EOF analysis cannot handle NaNs. We described this in Supplementary Result 2.3:

“Because Empirical Orthogonal Function (EOF) cannot deal with missing data and because high-frequency and seasonal variability is not the scope of this study, EOF was run on monthly non-seasonal SLA fields. However, some gaps corresponding to ice-covered areas (>15% in sea ice concentration) remained in the MSLA fields and have been systematically filled with zeros. Thus, the EOF results of the study area do not consider the winter variability associated with high latitude ice-covered areas (i.e. the northern Barents Sea, Kara Sea and the Greenland Sea). This is why the quality of the EOF in the northern and eastern parts of the EAC is quite poor due to low data coverage and is manually masked. However, this issue does not alter the Atlantic Water current monitoring in this study since the Atlantic domain is constantly ice-free.”

2) Concerning data coverage due to satellite availability. We insisted that from 1993 to 2016, the time-series can rely on several other altimeters with large inclinations: ERS-1 (1993-1995), ERS-2 (1995-2011), GFO (2000-2008, latitudes < 72degN), EnviSat (2002-2012), HY-2A (2011-ongoing), SARAL/Altika (2013-ongoing), CryoSat-2 (2011-ongoing), Sentinel-3A/3B (2016, ongoing). See Table below for more details. The lower amount of satellites at high latitudes above 66degN (and thus of data) would be an issue for resolving small scale features (e.g. finer than mesoscale or small mesoscale) but Volkov and colleagues concluded that it is consistent for synoptic and large mesoscale studies like this one (see also above answer to reviewer #4 and data coverage in Figure 2 from Volkov et al. 2012) using only ERS and Envisat satellites (1993-2012). Also, 80degN is the northern limit of our study area but virtual particles in our experiments rarely exceed 76degN in the Barents Sea. The detection limit of most satellites of 81.5degN is by far enough for this analysis. We now better highlighted in the main text, and more especially in the material and method, the recent publications that support that view, as well as discussing the strengths and the improvements of the dataset compared to the previous ones (e.g. reprocessing, use of CryoSat-2, Sentinel-3A after 2011).

Note here a resume of the Altimeters missions' characteristics (Table 9 of product documentation: <http://resources.marine.copernicus.eu/documents/QUID/CMEMS-SL-QUID-008-032-062.pdf>)

Altimeter mission	Cycle duration (days)	Latitude range (°N)	Number of track in the cycle	Inter-track distance at equator (km)	Sun-synchronous	Dual-frequency Altimeter	Radiometer on board	input data availability Start-End dates
Sentinel-3A	27	±81.5	770	~100	Yes	Yes	Yes	2016/12/13 (cycle 12) Ongoing
Jason-3	10	±66	254	~315	No	Yes	Yes	2016/02/17 (cycle 1) Ongoing
Jason-2	10	±66	254	~315	No	Yes	Yes	2008/07/12 (cycle1) 2016/10/02 (cycle 303)
Jason-2 Interleaved	10	±66	254	~315				2016/10/13 (cycle 305) Ongoing
Cryosat-2	29 (sub cycle)	±88	840	~98	No	No	No	2011/01/01 (cycle 13) Ongoing
Saral/AltiKa	35	±81.5	1002	~80	Yes	No	Yes	2013/03/14 (cycle 1) 2016/07/04 (cycle 35)
SARAL-DP/AltiKa	?	±81.5	?	-				2016/07/04 (cycle 100) Ongoing
HaiYang-2A	14	±81	386	~210	Yes	Yes	Yes	2011/10/01 (cycle 1) 2016/05/03 (cycle 120)
HaiYang-2A geodetic	168	±81	-	-				2016/03/24 (cycle 1) Ongoing
Topex/Poseidon	10	±66	254	~315	No	Yes	Yes	1992/09/25 (cycle 1) 2002/08/21 (cycle 365)
Topex/Poseidon Interleaved	10	±66	254	~315				23/08/2002 (cycle 366) 2005/10/08 (cycle 481)
Jason-1	10	±66	254	~315	No	Yes	Yes	2002/01/15 (cycle 1) 2009/01/26 (cycle 259)
Jason-1 Interleaved	10	±66	254	~315				2009/02/10 (cycle 262) 2012/03/03 (cycle 374)
Jason-1 Geodetic	10.91	±66	280	-				2012/05/07 (cycle 500) 2013/06/21 (cycle 537)
Envisat	35	±81.5	1002	~80	Yes	Yes (S-band lost after cycle 65)	Yes	2002/04/10 (cycle 5) 2010/10/18 (cycle 93)
Envisat-New	30	±81.5	862	-				2010/11/27 (cycle 96) 2012/04/08 (cycle 113)
ERS-1	35	±81.5	1002	~80	Yes	Yes	Yes	1992/10/23 (cycle 15) 1993/12/20 (cycle 27) And 1995/03/240 (cycle 41) 1996/06/02 (cycle 53)
ERS-1 geodetic	168	±81.5	-	-				04/10/1994 (cycle 30) 03/21/1995 (cycle 40)
ERS-2	35	±81.5	1002	~80	Yes	Yes	Yes	1995/05/15 (cycle 1) 2011/07/04 (cycle 169)
Geosat Follow On	17	±72	488	~165	No	No	Yes	2000/01/07 (cycle 37) 2008/09/07 (cycle 222)

Here is an extract of the Material section about altimetry data:

“

Altimetry.

The satellite altimetry product was generated by the Sea Level Thematic Assembly Center (SL-TAC) and distributed by the Copernicus Marine and Environment Monitoring Service (CMEMS, <http://marine.copernicus.eu/>). This is a daily gridded global product in delayed-time (re-analysis product ID: SEALEVEL_GLO_PHY_L4_REP_OBSERVATIONS_008_047) with a spatial resolution of 1/4° available from 1993 to present time. A 1/4° Mercator projection grid roughly corresponds to a resolution of 27.8 km in latitude x 9.5 km in longitude at 70°N. This product is generated using Sentinel 3A, Jason 3, HY2A, Saral/AltiKa, Cryosat 2, OSTM/Jason 2, Jason-1, Topex/Poseidon, Envisat, GFO, ERS-1/2. The SL-TAC

produced this data set with a special care for synchronization, homogenization, cross-calibration, correction of large-scale biases and cross-validation. Before merging missions by optimal interpolation, the SL-TAC filtered the data set to remove residual noise. The reader is referred to the product documentation for more details. The satellite altimetry product provides Sea Level Anomaly (SLA) that estimates the variation of the Sea Surface height around a vertical reference (Mean Sea Surface). The product also provides Absolute Dynamic Topography ($ADT = SLA + \text{Mean Dynamic Topography MDT}$), geostrophic velocity anomalies and absolute geostrophic velocities (derived respectively from SLA and ADT). The MDT employed by the TAC was the MDT-CNES-CLS13 produced by the Collecte Localisation Satellites (CLS) Space Oceanography division and distributed separately by Aviso (<http://www.aviso.altimetry.fr>).

SLA has been successfully used over the last 25 years to study the variability of sea level and surface geostrophic circulation at different spatial and temporal scales. However, most of the validation studies have focused on the areas below the polar circle ($< 66^\circ\text{N}$) such as in the Norwegian Sea where derived geostrophic currents were successfully evaluated against in situ current meters⁷⁰. The use of altimetry in Arctic regions has been avoided in the past due to (1) persistent sea ice cover and (2) because some radar altimeters (TOPEX/POSEIDON, Jason-1, OSTM/Jason-2 and Jason-3 missions) do not acquire measurements above the polar circle. First, sea ice was not an issue for this study because virtual particles were only evolving in the year-round ice-free Atlantic domain. For the trend and EOF calculation, data associated with sea ice were systematically precluded (see below). Second, measurements above 66°N can rely on several other satellites: ERS-1 and -2, Envisat, SARAL/AltiKa, GFO ($< 72^\circ\text{N}$), HY-2A, ICESat, Cryosat-2, Sentinel-3. A comprehensive validation of the gridded product of geostrophic velocities only from ERS-1/2 and Envisat satellites using surface drifters and tide gauges in the whole EAC, with a particular focus on the Barents Sea, was recently provided^{71,72}. The authors demonstrated that RMS differences between the drifter (corrected for Ekman currents) and altimetry velocities range within $7\text{-}15 \text{ cm s}^{-1}$ which is comparable to previous estimates at lower latitudes. More importantly, they also concluded that the data set provides sufficient spatial and temporal coverage to consistently resolve the synoptic and large mesoscale variability. Moreover, the product was recently found appropriate to monitor the long term northward Atlantic inflow in the Norwegian Sea⁴² with even lower RMS differences between drifters and satellites velocities ($0.7\text{-}8 \text{ cm s}^{-1}$).

The dataset used in this study was enhanced from additional satellites (i.e. CryoSat-2 and Sentinel-3A) that offer a better data coverage at high latitudes for recent years (since 2011). Surface currents inferred from altimetry were found to reflect the variability of the deeper layers in the slope current in the Nordic Seas⁴². For this study, the derived surface geostrophic velocities were evaluated again toward five long-term in situ current meters in the southwestern Barents Sea and showed reasonably good Pearson's correlation coefficients ($r = 0.43\text{-}0.71$, Supplementary Result 2.1). In addition, by evaluating the effect of steric and mass related effect on the SLA, we demonstrated that the increasing long-term trend was driven by change in currents (Supplementary Result 2.2).

”

We hope that those improvements will satisfy you.

A point that is not mentioned in the methodology is how geostrophic velocities are computed. As showed in Liu Y., Weisberg R. H., Vignudelli S., Roblou L., Merz C. R.: Comparison of

the X-TRACK Altimetry Estimated Currents with Moored ADCP and HF radar Observations on the West Florida Shelf, in Special Issue “COSPAR Symposium”, Journal of Advances in Space Research, 50 (8), 1085-1098, doi:10.1016/j.asr.2011.09.012, 2012, white noise in altimetry data can affect the accuracy of the slope estimate from SLAs. Are the authors using a filter ? I am stressing about providing details about processing because we need to ensure that other scientists can reproduce the work

We used the ‘ready-to-use’ surface absolute geostrophic velocities directly provided and freely available on Copernicus website (dataset: SEALEVEL_GLO_PHY_L4_REP_OBSERVATIONS_008_047). This should make reproducibility of our work very straightforward. The SL-TAC/Copernicus SLA product are smoothed before calculation of the gridded SLA and of the geostrophic velocities using a low pass filter applied on the along track signal. Namely, the Copernicus documentation says: *“II.4.6.2- Along track (L3) noise filtering: The filtering processing consists in removing from along-track measurements the noise signal and short wavelength affected by this noise. This processing consists in a low-pass filtering with a cut-off wavelength of 65km over the global ocean. This cut-off wavelength comes from the study by Dufau et al. (2016) and is discussed in Pujol et al (2016).”*

Note as well that the SL-TAC standard processing used the MDT CLS 13 at the time we did this analysis. However, we just learned that SL-TAC updated their processing last month (December 2019) with the new MDT CLS 2018 (we insist that this study use MDT-CLS-13 not 18). This is completely out of our control. In any case, anyone downloading this dataset will have exactly the same SLA fields that we have used in this study without any additional post-processing from our side. They can use the MDT-CLS13 to reconstruct ADT fields, and derive geostrophic velocities. The only ‘processing’ was the construction of 30-days composites for the EOF and trend analysis. We will make sure to provide all necessary codes and data (e.g. monthly fields) in a public repository in case of publication. The New MDT CLS 2018 has finer resolution and sharper gradients which should in principle increase satellite velocities and reduce errors (usually satellite velocities are underestimating the ‘true’ velocities from drifters). The underestimation of geostrophic velocities could be also due to the important ‘smoothing’ (65km filtering) done by the SL-TAC.

Concerning the validation, the authors just point to references, in the manuscript. In my opinion, the authors have to provide some quantitative figures of the validation results from bibliography (Norwegian Sea, the whole EAC, Barents Sea) and from their independent validation analysis against long-term in situ measurements in the southeastern Barents Sea. The readers will be then pointed to the Supplementary Material to better see in detail where the validations were made, which altimeter data were used, which in situ data were used. The reader also expects to have here a discussion of the statistics from the comparisons between altimetry and in situ data. I agree to mention Volkov (2012) who is not validating velocities but sea level anomaly, however, these figures will be supportive of the new product (that is certainly better than previous one). Ray et al (2018) is another important reference as they use the same product and the comparison is about altimeter-derived velocities against those measured by drifters.

We entirely concur with the reviewer that validation is important and that we should better value the validation efforts that have been done in previous study or in this study by expressing quantitative results.

First, we would like to note that there are different ways to consider validation. We believe that Volkov et al., 2012 did provide an important validation of satellite-derived velocities, and not only sea-level anomalies. For instance, they did a thorough comparison of satellite altimetry velocities with the most complete data set of surface drifters trajectories (as for Raj et al. 2018 for the Norwegian Sea). They concluded that satellite derived geostrophic velocities are consistent with the mesoscale activity and associated errors are comparable with the World Ocean. For reference, we copied below Figures. 10 and 11 from Volkov et al. 2012 which compare satellite derived geostrophic velocities with drifters:

Figure 10. The time mean U and V components of the (top) drifter and (middle) altimetry velocities averaged over $1^\circ \times 1^\circ$ bins, and (bottom) the difference between them. The drifter velocities were corrected for Ekman currents. The altimetry data were interpolated at the trajectories of drifters.

Figure 11. The RMS of the differences (cm/s) between the altimetry and drifter (a) U and (b) V velocity components computed over $1^\circ \times 1^\circ$ bins. The altimetry data were interpolated at the trajectories of drifters. The drifter velocities were corrected for Ekman currents.

We acknowledge they used an older version compared to what we use. However, the same analysis as presented in Volkov et al. has been repeated at global scale on the most recent version of the satellite velocity product we use. This is presented in the CMEMS documentation and the result can be found at this link: <http://resources.marine.copernicus.eu/documents/QUID/CMEMS-SL-QUID-008-032-062.pdf>. We below copied the key figure reproducing Volkov et al. analysis at global scale. Clearly, the Barents Sea shows encouraging and very reasonable low to moderate errors for the time period 1993-2017 comparable to the world Ocean.

Figure 18: Zonal (left) and meridional (right) RMS of the difference between DUACS geostrophic current and drifter's measurements over the period [1993-2017] (units: cm/s).

In the new manuscript, we have now dedicated an entire paragraph to explicitly and quantitatively describe the validation efforts in the material and method of the main text in the Altimetry section (see below). After describing the existing validations from Volkov and colleagues and Raj and Colleagues, (We add the range of RMS difference between drifter and altimetry velocities), we present the Pearson's correlation coefficient from our own validation with the 5 in situ moorings in the southwest Barents Sea, referring to the Supplementary information for more details. See below the related two last paragraphs of the Altimetry Material section:

“[...] SLA has been successfully used over the last 25 years to study the variability of sea level and surface geostrophic circulation at different spatial and temporal scales. However, most of the validation studies have focused on the areas below the polar circle (< 66°N) such as in the Norwegian Sea where derived geostrophic currents were successfully evaluated against in situ current meters⁷⁰. The use of altimetry in arctic regions has been avoided in the past due to (1) persistent sea ice cover and (2) because some radar altimeters (TOPEX/POSEIDON, Jason-1, OSTM/Jason-2 and Jason-3 missions) do not acquire measurements above the polar circle. First, sea ice was not an issue for this study because virtual particles were only evolving in the year-round ice-free Atlantic domain. For the trend and EOF calculation, data associated with sea ice were systematically precluded as stated before. Second, measurements above 66°N can rely on several other satellites: ERS-1 and -2, Envisat, SARAL/AltiKa, GFO (<72°N), ICESat, Cryosat-2, Sentinel-3. More importantly, a comprehensive validation of the gridded product of geostrophic velocities only from ERS-1/2 and Envisat satellites using surface drifters and tide gauges in the whole EAC, with a particular focus on the Barents Sea, was recently provided^{71,72}. The authors demonstrated that RMS differences between the drifter (corrected for Ekman currents) and altimetry velocities range within 7-15 cm s⁻¹ which is comparable to previous estimates at lower latitudes. They also concluded that the data set provides sufficient spatial and temporal coverage to consistently resolve the synoptic and large mesoscale variability. Moreover, the product was recently found appropriate to monitor the long term northward Atlantic inflow in the Norwegian Sea⁴² with even lower RMS differences between drifters and satellites velocities (0.7-8 cm s⁻¹).

The dataset used in this study was enhanced from additional satellites (i.e. CryoSat-2 and Sentinel-3A) that offer a better data coverage at high latitudes for recent years (since 2012). Surface currents inferred from altimetry were found to reflect the variability of the deeper layers in the slope current in the Nordic Seas⁴². For this study, the derived surface geostrophic velocities were evaluated again toward five long-term in situ current meters in the southeastern Barents Sea and showed reasonably good Pearson's correlation coefficients ($r = 0.43-0.71$, Supplementary Result 2.1). In addition, by evaluating the effect of steric and mass related effect on the SLA, we demonstrated that the increasing long-term trend was driven by change in currents (Supplementary Result 2.2).

”

The authors also added a validation against current meter measurements from 50 m depth in the Barents Sea Opening (section 2.1. Validation of satellite-derived geostrophic velocities toward in situ instruments at BSO in supplementary material). In situ data are averaged daily to remove tides and then averaged furtherly weekly. A 4-weeks running mean is also applied to the time series. One plot shows the comparison at moorings with statistics and another one is showing velocity trends. Here there are some points that need to be clarified: 1) a map showing the mooring locations is necessary; 2) which altimetry grid point is selected for

comparison (closest? radius?); 3) altimeter-derived velocities are daily (Is the running mean applied to this data set? If not, why?).

- 1) Absolutely, moorings are now shown in the map in Fig.S1 provided in the section 1.1 of the SI.
- 2) This is true - we did not provide information about how we extracted the geostrophic velocity fields to compare with moorings. We averaged pixels overlapped by a 0.1-degree radius. The moorings are located at about 19.5 degE and every 0.5deg from 70 to 74.5 degN. The mooring positions always fall exactly between 2 pixels of the CMEMS product in longitude and latitude. A 0.1-degree radius around the position of the mooring therefore corresponds to averaging of a block of 4 pixels around it. We now explicitly explain this procedure in the section 2.1:
- 3) Absolutely, we applied the exact same processing as for the mooring (4-weeks running mean). We added this information in the SI.

To include the new information, the first paragraph of Supplementary Material's section 2.1 becomes:

“To investigate the reliability of the satellite altimetry-derived velocity field, the satellite-derived geostrophic velocities were compared with in situ current meter data at 50 m depth from moorings between 71°30 and 73°30'N at the Barents Sea Opening (BSO, see in Fig S1, time-series updated from⁴). Note that data from moorings are daily averaged to remove diurnal and semidiurnal tidal cycle, then weekly (7-days) averaged. Following the same method, 7-days composites maps from satellite altimetry data were produced, and time series were extracted averaging pixels within a 0.1° radius around the mooring location. In addition, a 4-weeks running mean were applied to both time series before deriving the statistics. The time series showed relatively similar varying current speed ($r=0.43-0.71$) and direction (Fig. S3). Overall, the geostrophic velocities derived from altimetry were generally underestimated in agreement with previous estimations⁵.”

The authors will clarify differences between altimetry-derived and in situ data velocities in the answers to the reviewers. Ocean velocity fields inferred from altimetry exhibits various uncertainties. The altimeter measures both the barotropic and baroclinic signals. Altimetry provides an integrated measurement of the water column. Sea level is transformed in velocity using the geostrophic approximation. In order to avoid aliasing, high frequency wind effects have to be removed too. Current meters measure pointwise real velocities. Disagreements can be explained in terms small-scale variability, ageostrophic behaviour, increased variance going closer coastline. There are also several errors related to the altimetry system (error budget is at cm level in open ocean). By adding the Ekman velocity component to altimeter-derived velocity, the agreement is much better (see e.g. Liu et al. (2012) for a thorough comparison). This reference can be cited to reinforce the potential of altimetry to derive ocean velocity fields.

Thank you for this advice. We added that using already available products that provide Ekman velocities (and now Stokes drift), we could further improve the precision the satellite ocean currents estimations (we have referred to Liu et al. 2012). We added this information at the end of section 2.1 in the SI:

“Note that recently, CMEMS is distributing a dataset produced by the Centre National d'Études Spatiales (CNES) and the Collecte Localisation Satellites (CLS) in the framework of the Globcurrent ESA project (<http://www.globcurrent.org/>) that deliver satellite-derived geostrophic velocities together with modelled Ekman transports and Stokes drift. The consideration of the Ekman and Stokes components could further improve the estimation of ocean currents and reduce errors²⁰ and should be considered for further studies. Alternatively, the European Space Agency Climate Change Initiative (CCI) project on “Sea Level” has reprocessed altimeter data to provide satellite-based sea level products to study climate-scale variations of sea level globally and in the coastal zone²¹. The global product (available via ESA) provides monthly means of Mean Sea Level Anomaly 1/4° during the 1993-2015 period using TOPEX/Poseidon, Jason-1, Jason-2, ERS-1, ERS-2, GeoSat Follow-On (GFO), Envisat, SARAL/AltiKa and CryoSat-2 with special care of editing, cross-calibration, homogeneous corrections, removal of global and regional biases, homogenization of long-spatial-scale errors, monthly optimal interpolation gridding. The need for daily temporal resolution for the Lagrangian experiments hindered us to use this data set.”

As stated by authors, maps of geostrophic velocity error do not exist yet. They used two complementary approaches to assess the robustness of the Lagrangian technique to the presence of these errors. I think that they did all their best to follow reviewer's recommendation.

I have an additional question about the trends mentioned in the manuscript. In particular I would like the authors considering to perform a significance test of the estimated trends that is missing in the actual version.

We already did a t-test for the calculation of the linear trends in the first version of the manuscript. We decided to only show (>90%) significant parts of the trends in Fig. S9 ($p < 0.1$). But to avoid confusion, we now added a second panel to explicitly show the map of p -values as requested. Please, see changes in new Figure S9.

In summary, concerning the satellite altimetry observations, the authors have made a great effort to follow reviewer's recommendation. Before publishing, there is a need to better quantify the quality of the data set and some numbers about accuracy need to be provided in the main manuscript (a synthesis from the literature and comparison showed in the supplementary section). In the present for, there are only generic sentences. More quantitative results need to be added.

Thank you so much for raising this, we are also convinced that more supporting facts are necessary in the main text. We have re-written extensively the associated data and method sections in the main text, and improved the Supplementary material. Please, find all modifications in the tracked change manuscript and see the changes documented above.

Finally, about SST and ocean color data sets, the authors use standard products available from providers. As for altimetry, the authors have to comment the quality and accuracy of SST data in the area of investigation. I am not here to ask more analysis, but a synthesis from the bibliography about the confidence in using these data sets is particularly necessary for the broad readership of this journal.

We agree this is an important issue. The product used in this study is the following: <https://www.esrl.noaa.gov/psd/data/gridded/data.noaa.oisst.v2.highres.html>. A documentation can be found here: <https://www.ncdc.noaa.gov/oisst>.

The SST product is freely available, daily on a 1/4 degree grid from 1981 to present time. Again, it is merged product combining different satellites and in situ platform to reach a maximum data coverage. We did process the SST in the same way than altimetry, building composite maps avoiding errors >50%, ending up with 100% data coverage on monthly fields for the concern area (8degW-60degE / 50-76degN). One caveat of the data set is the presence of large errors close to the ice edge. But in this study, we only used the 4degC isoline which was not affected by the sea ice edge generally associated with a temperature range between -2 and 0degC. We have also completely updated and improved the data section in the main text about SST:

“Sea Surface Temperature.

We used the NOAA 1/4° degree and daily Optimum Interpolation Sea Surface Temperature version 2 (or daily OISSTv2, provided by the NOAA/OAR/ESRL PSD, Boulder, Colorado, USA, <https://www.esrl.noaa.gov/psd/>⁷³) to extract the 4°C isotherms during the month of March. This dataset is constructed by combining satellites observations from the Advanced Very High Resolution Radiometer (AVHRR) and the Advanced Microwave Scanning Radiometer on the Earth Observing System (AMSR-E) with in situ platforms (ships, buoys) on a regular global grid. Spatially complete SST maps are generated by interpolation to fill the gaps. The dataset is delivered with error fields providing a measure of confidence allowing us to build monthly composites maps excluding values with estimated errors higher than 50%, following the same method than for the altimetry dataset. Note that the dataset also provides daily sea ice concentration maps which were used as masks for the altimetry data set.”

We applied the same procedure for Ocean Color data in the data section of the main article:

“Ocean color remotely sensed Particulate Inorganic Carbon (PIC).

Ocean colour satellites can detect coccolithophore blooms because they produce calcite plates (coccoliths) that shed light and produce ‘turquoise’ colored waters. The remotely sensed information offers a 20-year time series between 1998 and today. We used the GlobColour database (<http://hermes.acri.fr>) which provides a continuous data set of ocean color satellite data products from the merging of 5 ocean color sensors (MEdium Resolution Imaging Spectrometer (MERIS), Moderate Resolution Imaging Spectroradiometer (MODIS), Sea-viewing Wide Field-of-view Sensor (Sea-WiFS), Visible Infrared Imaging Radiometer Suite (VIIRS) and Ocean and Land Colour Imager (OLCI-A)). These sensors measure visible and near-infrared solar radiation emerging from the ocean surface layer. Such remotely-sensed information is only available during the daytime and in the absence of ice and clouds.

Level 3 PIC products were derived from NASA’s standard PIC algorithm^{74,75}. This algorithm is based on a robust relationship between the light backscattering coefficient and the concentration of coccoliths, calcite plates forming the coccosphere of the coccolithophores^{76,77}. Some coccolithophore species, including Ehux, produce and release coccoliths into the water during the later stages of a bloom⁷⁸⁻⁸⁰, creating large patches of

highly reflective waters, which can be observed from space⁵⁷. Satellite-derived PIC concentration provides a good proxy for Ehux coccolith concentration because Ehux dominates the coccolithophore population in the Norwegian Sea⁸¹ and in the Barents Sea⁸². Moreover, satellite PIC products have been successfully validated during Ehux blooms in the Barents Sea^{18,82,83}, as well as in the global ocean^{57,75,84}.”

I am available to have a check of the revised version.

We very much thank the reviewer for the thorough reading, and insightful comments that, we believe, improved the clarity of our work.

Minor comment

Please provide reference of Cipollini et al. 2014.

Thank you. This is now done in the SI when introducing the alternative ESA-CCI data set.

We have done some minor additional modifications. There are visible in the tracked change manuscript but we have tried as much as possible to list them below:

- We improved Figure 3. In Figure 3 caption, we realized we did not specify that snapshots are from the extremums of the time-series which correspond to the months of December but that the trend considers all months over the 1993-2016 period. We also did not distinguish between 0-values and No-Data which were both white (now respectively in white and dark gray). We also spotted an issue with the sea ice mask in Figure 3. All those issues are now corrected along with the corresponding Figure S8 in the SI.
- Same story for Figure 4, the color code did not allow to distinguish 0-values from No-Data. We changed the color code with a dark background for no-data to resolve this issue as for Fig. 3.
- We have brought some minor improvements (wording) in the method section concerning the Lagrangian technique et the leading-edge detection.
- In the SI, we added the missing information that for the leading-edge analysis, the summer maps of PIC were derived as the maximum concentration of July – August – September following the method from Neukermans et al. 2018.
- Note that method section about the EOF has benefited from input from the former Supplementary information. We do say more explicitly that we have removed the seasonal cycle and how we built monthly composites.
- Line 214, we replaced “Atlantic Waters” by “Atlantic ecosystems”
- Every time we talked about Ehux “distribution”, we specified “spatial distribution” to be clearer.

Best regards,
On behalf of all co-authors,
Laurent Oziel

REVIEWERS' COMMENTS:

Reviewer #5 (Remarks to the Author):

I have read carefully the revised manuscript (including supplementary material) and the answers to reviewers. I appreciate the efforts of the authors to improve the original manuscript. The revised material is now much improved with reference to the structural aspects and scientific content. It is also better readable for audience, with more details and explanations. I am satisfied how authors took care of my comments. In particular, the authors always provided elaborated answers point by point. It was easy to follow their rationale in answering and how text was changed. They have better described all satellite data sets (altimetry, SST and ocean color) in the section "Material and methods" of the main text as well as in the supplementary data sections 1.1 and 1.2. Now there are more quantitative results from the statistical analysis with error/uncertainties that better support the scientific interpretation. The authors also reinforced the validation of satellite data sets with direct analysis and bibliographic support.

Overall, I agree with the other reviewers that this is an important study showing the link between changes in surface currents and marine species. It contributes to advance the scientific knowledge of the Arctic region. This study also demonstrates the important support of satellite data sets to understand scientific questions in the Arctic region. Having said that, I feel that the manuscript is now acceptable for publication.

RESPONSE TO REVIEWERS COMMENTS:

Reviewer #5 (Remarks to the Author):

I have read carefully the revised manuscript (including supplementary material) and the answers to reviewers. I appreciate the efforts of the authors to improve the original manuscript. The revised material is now much improved with reference to the structural aspects and scientific content. It is also better readable for audience, with more details and explanations. I am satisfied how authors took care of my comments. In particular, the authors always provided elaborated answers point by point. It was easy to follow their rationale in answering and how text was changed. They have better described all satellite data sets (altimetry, SST and ocean color) in the section "Material and methods" of the main text as well as in the supplementary data sections 1.1 and 1.2. Now there are more quantitative results from the statistical analysis with error/uncertainties that better support the scientific interpretation. The authors also reinforced the validation of satellite data sets with direct analysis and bibliographic support.

Overall, I agree with the other reviewers that this is an important study showing the link between changes in surface currents and marine species. It contributes to advance the scientific knowledge of the Arctic region. This study also demonstrates the important support of satellite data sets to understand scientific questions in the Arctic region. Having said that, I feel that the manuscript is now acceptable for publication.

We thank the reviewer #5 for the final remarks and for helping in improving the quality of the manuscript.